# Uncovering the Hidden Numbers of Nature in the Standard Accounts of Society: Application to a Case Study of Oak Woodland *dehesa* and Conifer Forest Farms in Andalusia-Spain

**Pablo Campos \*, Bruno Mesa and Alejandro Álvarez**

Spanish National Research Council (CSIC), Institute for Public Goods and Policies (IPP), C/Albasanz, 26–28, E-28037 Madrid, Spain; bruno.mesa@cchs.csic.es (B.M.); alejandro.alvarez@cchs.csic.es (A.Á.)

\* Correspondence: pablo.campos@csic.es; Tel.: +34-91-602-2535

**Abstract:** The standard System of National Accounts (SNA) does not estimate the margins of the products without market prices consumed because it assumes that the cost prices of the final products consumed correspond to the consumer marginal willingness to pay (MWTP). Valuations of products consumed without market prices at their cost prices may not coincide with their simulated exchange values (SEV) that would be paid by consumers. This inconsistent SNA valuation can be avoided by simulating stated or revealed market prices based on consumers' demands. Our Agroforestry Accounting System (AAS) methodology estimates the margins of the individual products without market prices based on the consumer MWTP. The SEV of private owners and public consumers MWTP for these non-market products are estimated in this study by applying stated and revealed preference valuation methods. The objectives of this study are to compare the environmental incomes, ecosystem services and profitability rates obtained by applying the AAS and the refined SNA (rSNA) methodologies to the case-study oak woodland *dehesa* and conifer forest farms in Andalusia, Spain. The 41 farms comprise 26 large oak woodland *dehesa* farms in which trees of the *Quercus* genus predominate, and 15 conifer forest farms where *Pinus* species predominate. In the studied farms, 20 individual activities have been identified which 19 are common to both the AAS and rSNA approaches, along with the additional activity of carbon which is registered in the AAS. Ownership rights of 13 private activities correspond to the farmer and 7 public activities to the government. In 2010, the case-study results show that livestock and game species consume grazed fodder which represents 50% and 95%, respectively, of their total forage units consumed in the period 2010. Livestock farming accounts for 31% of the labour compensation in the private oak woodland *dehesa* farms and 1% in the public conifer forest farms for the farm activities as a whole. The ecosystem services measured by the AAS in the privately-owned oak woodland *dehesa* and publicly-owned conifer forest farms are 2.7 and 4.6 times greater, respectively, than those estimated by the rSNA. The environmental incomes measured by the AAS for the privately-owned oak woodland *dehesa* and publicly-owned conifer forest farms account for 61% and 53%, respectively, of their total incomes.

**Keywords:** System of National Accounts; Agroforestry Accounting System; ecosystem services; environmental asset; total income

## 1. Introduction

In the last decade, the United Nations Statistics Division (UNSD) has been debating over the System of Environmental-Economic Accounting—Ecosystem Accounting (SEEA EA) as an internationally recognized statistical principles and recommendations for valuation of ecosystem services and environmental assets of the Nature-based economic activities [1].

The standard System of National Accounts (SNA) limits the measurement of the market final products consumed at basic prices to the wood products (timber, cork, firewood) and non-wood forest products (resin, industrial wild fruits, hunting, livestock market products, and other minor products) harvested in the period along with own-account manufactured fixed investments in forestry improvement (e.g., plantation), construction and equipment in forest areas at national/sub-national scale [2–5]. The basic prices are obtained by adding the government compensations (operating subsidies net of taxes on products) to the estimated values at producer prices (market prices). It is accepted that the oak woodland *dehesa* (henceforth *dehesa*) and conifer forest (henceforth forest) farms provide the owners and society as a whole with non-commercial goods and services which are not registered in accordance with their real value (a value of zero for the net operating margins is assumed) or are completely omitted in the standard SNA. Although the territorial unit for the application of the SNA is national/sub-national, the economic concepts can also be applied at individual farm scale.

Our Agroforestry Accounting System (AAS) approach applied in this research extends the SNA to farm scale in order to tackle the challenge of incorporating the numbers of nature in the total sustainable income at social prices of the individual activities of the case study farms. We define the social prices as the unit values of the final products consumed paid in an observed market transaction or, in the absence of a market, the simulated exchange values (excluding consumer surpluses) of the final products consumed without market prices according to their marginal willingness to pay (MWTP) stated or revealed by the consumers. This AAS research develops recent advances in the valuation methods of simulated transaction prices for final products without market prices consumed [6,7].

Previous to this study, we have published results for case studies of farms in which the SNA and AAS methodologies are applied at silvopastoral farm scale in areas with a Mediterranean climate, namely, California, Spain, France, Portugal and Tunisia [6–18]. We defined environmental income at environmental prices as the total nature contribution to total income accruing from farm activities net value added and capital gain. We define the environmental price as the unit resource rent. The letter is defined as the depletion (extractions of natural resource in the period) plus operating return of environmental fixed asset [1].

In the case studies in this research we are interested in comparing the economic valuations under the AAS and our refined standard System of National Accounts (rSNA) in the large *dehesa* and forest farms with livestock and game species grazing, belonging to non-industrial owners. A non-industrial owner is characterized by the voluntary acceptance of commercial operating margins at basic prices opportunity cost for their manufactured investments below an assumed baseline competitive one in exchange for greater self-consumption of private amenity services (individual private owner) or donation to third parties of non-commercial intermediate products of services which favour the production of public products (private or public institutional owners). These non-business as usual behaviours are explained by the fact that the non-industrial land and livestock owners are assured, in exchange, greater self-consumption of private amenities. The literature reviewed shows that the owners of the large Spanish *dehesa*, Portuguese *montado* and Californian ranch farms with woodland of the *Quercus* genus, as well as those of *Pinus* forest farms in Corsica and Andalusia, accept market profitability rate from the commercial economic activities below that which would be obtained by selling their farms and investing the financial capital in other alternative non-agrarian assets.

In this study, the AAS and slightly refined SNA (henceforth rSNA) are applied in Andalusia, Spain, to a group of *dehesa* farms with a predominance of *Quercus* species, and forest farms where species of the *Pinus* genus predominate. It is assumed that the case-study farms in this research present the economic rationales and trends of the

non-industrial private and public owners of large silvopastoral farms with a predominance of the *Quercus* and *Pinus* genera.

The data for the 2010 period in this study come from field work by the authors in the 41 large agrosilvopastoral *dehesa* and forest farms under the REnta y CApital de los Montes de ANdalucía (RECAMAN) project and other sources of information provided by the governments of Andalusia and Spain. Furthermore, the *ad hoc* surveys that we carried out as part of the RECAMAN project provide an additional source of data employed in the valuations at simulated transaction prices of the case-study farm products without market prices consumed [19–22].

The farmers manage 13 of the private activities: timber, cork, firewood, nuts, livestock grazing, conservation forestry, aromatic plants, hunting, commercial recreation, residential, livestock, agricultural crops and amenity. The government is the trustee of society for the 7 public activities: fire services, free access recreation, mushrooms, carbon, landscape, biodiversity and water.

The objective in this study focusses on comparing the *dehesa* and forest farms incomes, ecosystem service and profitability rates estimates under the AAS at social prices and the rSNA at basic prices for the individual activities, those corresponding to the farmer, the government and the aggregate for the case-study farm activities as a whole. This study compares the results obtained for the same variables under the AAS methodology when the products are valued at producer (market), basic or social prices.

The novelties of this research compared with that of [16] are that we present the measurements of values added, ecosystem services and profitability rates valued at social prices with the incorporation of non-commercial intermediate products of amenity (ISSnca) and donation (ISSncd) services for the individual activities and the aggregate activities of the farmer, the government and those of the case-study farm as a whole. The incorporation of the ISSnca/d service means that their counterparts of ordinary own non-commercial intermediate consumptions of services (SSncooa/d) are registered at the same time. Other novelties in this research include, on the one hand, the comparisons of the groups of public *dehesa* and private forest farms, and on the other, the comparison of the rSNA and AAS methodologies. These latter comparisons underline the sensitivity of *dehesa* and forest farms incomes to ownership and prices types.

This research continues in Section 2 with a brief summary of the biophysical and institutional characteristics of the case-study *dehesa* and forest farms. Section 3 describes the concepts and methods of the AAS and rSNA methodologies applied in the case study. Section 4 presents the main economic results for income and capital and highlights the contributions of the numbers of nature to the environmental incomes and ecosystem services of the farms. Section 5 discusses, on the one hand, the strengths and weaknesses of the results for the case-study farms and, on the other, highlights the implications for government policy of the SEEA EA recommendations for voluntary implementation of the measurement of ecosystem services and environmental assets by the offices for statistics and other government departments. Finally, Section 6 sums up the main results and approach advances of this research.

## 2. Oak Woodland *dehesa* and Conifer Forest Farms Case Study in Andalusia

In this section we describe vegetation cover, environment, and institutions settings of *dehesa* and forest farms case study in Andalusia.

### 2.1. Dehesa and Forest Farms Vegetation Cover

In the region of Andalusia, species of the *Quercus* and *Pinus* genera occupy an area of 38.6% and 20.3%, respectively, of the total area of *montes* (forests, woodlands, shrubland and permanent grassland) [20] (Table S1, p. 67). Among the species of the *Quercus* genus, holm oaks (*Quercus ilex* L.) and cork oaks (*Quercus suber* L.) make up 32.1% and 5.7%, respectively, of the total area of Andalusian *montes.* Of the *Pinus* species, the most

widespread in terms of area are *Pinus halepensis* Mill., *Pinus pinea* L., *Pinus pinaster* Ait. and *Pinus nigra* Arn., in that order.

Native tree species of the *Quercus* genera predominate in the case-study *dehesa* farms and *Pinus* species predominate in the forest farms. The trees and bushes of the *dehesa* farm has been thinned, favouring the areas occupied by *Quercus ilex* and *Quercus suber* with a tree canopy cover of less than 75% of the wooded area. Trees silvicultural treatments meanwhile, favour timber production given the predominance of *Pinus halepensis*, *Pinus pinea*, *Pinus pinaster* and *Pinus nigra* in the forest farms. Henceforth we will refer to the woodland farm with a predominance of areas occupied by *Quercus ilex* or *Quercus suber* as *dehesa* farm and we will distinguish between private and public *dehesa* farms. We refer forest farm as the one with a predominance of species of the *Pinus* genus, which we will also separate into private and public forest farms (Table 1, Figure 1).

**Table 1.** Vegetation cover and other land uses in the large *dehesa* and forest farms case study in Andalusia.

| Class | Privately-Owned *dehesa* Farm | | Publicly-Owned *dehesa* Farm | | Privately-Owned Forest Farm | | Publicly-Owned Forest Farm | |
|---|---|---|---|---|---|---|---|---|
| | ha | % | ha | % | ha | % | ha | % |
| 1. Useful agrarian land | 15,285 | 99 | 17,724 | 99 | 960 | 99 | 53,153 | 99 |
| Open woodland | 11,788 | 77 | 9661 | 54 | 165 | 17 | 2780 | 5 |
| *Quercus ilex* | 7138 | 46 | 3978 | 22 | 144 | 15 | 2654 | 5 |
| *Quercus suber* | 3058 | 20 | 3222 | 18 | 9 | 1 | 39 | 0 |
| Others oaks | 461 | 3 | 1664 | 9 | 7 | 1 | 20 | 0 |
| Wild olive | 1131 | 7 | 798 | 4 | 5 | 1 | 67 | 0 |
| Eucalyptus | 112 | 1 | | | 63 | 7 | 254 | 0 |
| Shrubland [1] | 1665 | 11 | 1887 | 11 | 8 | 1 | 7315 | 14 |
| Grassland | 485 | 3 | 37 | 0 | 3 | 0 | 1656 | 3 |
| Conifers | 687 | 4 | 5685 | 32 | 675 | 70 | 31,936 | 59 |
| *Pinus pinea* | 416 | 3 | 2598 | 15 | 651 | 67 | 3625 | 7 |
| *Pinus pinaster* | 139 | 1 | 2512 | 14 | 24 | 2 | 9240 | 17 |
| *Pinus nigra* | 1 | 0 | | | | | 4942 | 9 |
| *Pinus halepensis* | 92 | 1 | 9 | 0 | | | 6813 | 13 |
| *Pinus sylvestris* | | | | | | | 4782 | 9 |
| Others conifers | 39 | 0 | 566 | 3 | 0 | 0 | 2535 | 5 |
| Other forest [2] | 287 | 2 | 438 | 2 | 24 | 2 | 8330 | 15 |
| Agriculture | 262 | 2 | 15 | 0 | 22 | 2 | 883 | 2 |
| 2. Others [3] | 87 | 1 | 185 | 1 | 10 | 1 | 770 | 1 |
| 3. Total | 15,372 | 100 | 17,909 | 100 | 970 | 100 | 53,923 | 100 |

Notes: [1] Includes shrubland and mix shrubland-grassland. [2] Includes riparian forests, other species and mix oaks-conifers forests. [3] Infrastructure an unproductive surface.

In the private *dehesa* of the case study, species of the *Quercus* and *Pinus* genera cover 77% and 4%, respectively, of the total area (Table 1). Among the species of the *Quercus* genus, holm oaks and cork oaks make up 40% and 20%, respectively, of the total area. In the public *dehesa* the area occupied by *Quercus* drops to 54% and that occupied by *Pinus* rises to 32%. As regards the drop in *Quercus* species in the public *dehesa* compared to the private *dehesa*, the difference is greater in the case of holm oak, which falls to 22% while cork oak falls only slightly to 18% (Table 1).

In the case-study public forest, species of the *Pinus* and *Quercus* genera occupy 59% and 5%, respectively of the total area (Table 1). Of the *Pinus* species in the public forests *Pinus pinaster* and *Pinus halepensis* make up 17% and 13%, respectively, of the total area. In the case of the private forest, *Pinus* makes up 70% of the forested area and *Quercus* accounts for 17% (Table 1). *Pinus pinaster* and *Pinus pinea* make up 2% and 67%, respectively, of the total area of the private forest.

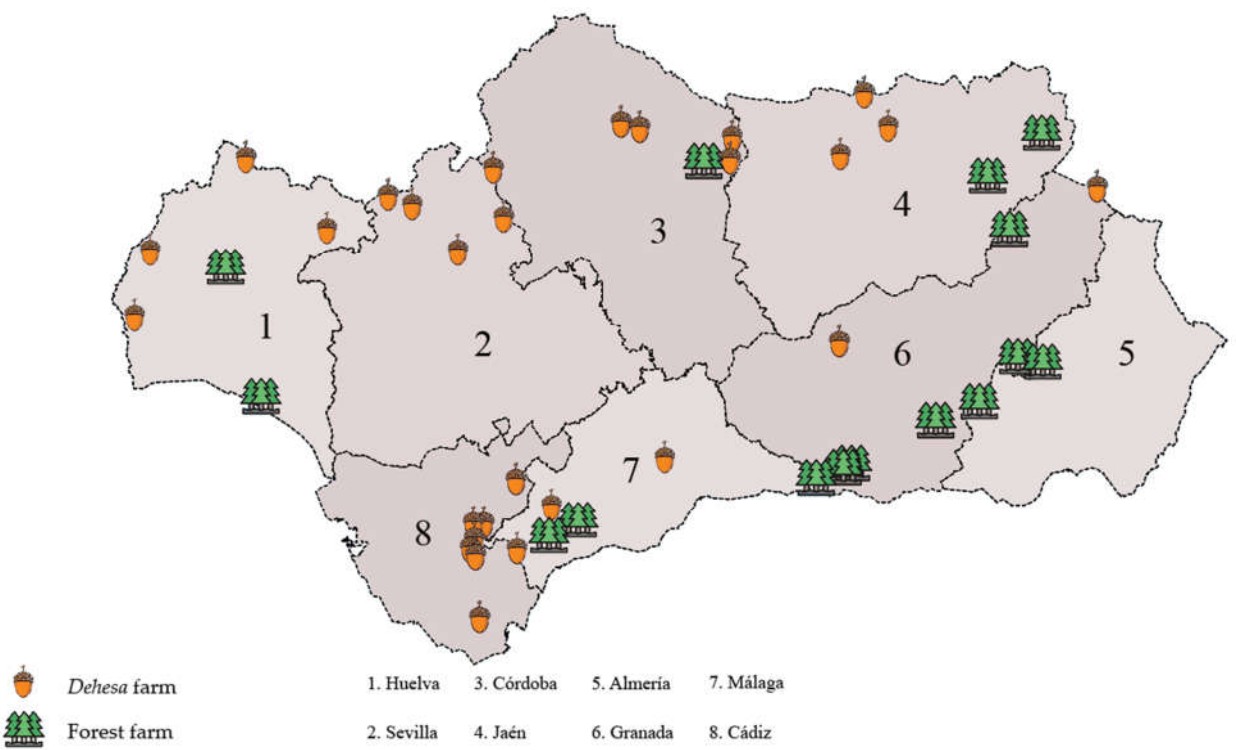

Dehesa farm
Forest farm

1. Huelva   3. Córdoba   5. Almería   7. Málaga
2. Sevilla   4. Jaén   6. Granada   8. Cádiz

**Figure 1.** Location of the case-study large *dehesa* and forest farms in Andalusia autonomous region of Spain.

The case-study *dehesa* and forest farms in this research also have small areas of agriculture, although the most important differences among the farms are those relating to the percentage occupation *Quercus* and *Pinus* species. This difference in the percentage contribution of each species among the group of 41 case-study farms and those for the region of Andalusia is one of the reasons why it is not possible to derive any significance in terms of statistical representativeness from the absolute aggregate physical and economic indicators beyond that exclusively relating to the group of case-study farms. However, in the qualitative analysis of the results, the predominant economic trends are considered to be significant for the *montes* (*dehesa* and forest farms) of Andalusia and by extension, in general for the Mediterranean *montes* of Spain.

### 2.2. Dehesa and Forest Farms Environment and Institutions

While the economic management of forests in temperate climates of the centre and north of Europe is driven by increased growth of timber, in Mediterranean regions, the main drivers in the management of the large *dehesa* and forest farms are grazing, recreational big game hunting, private amenity services, public recreational services, cultural landscape conservation services and threatened wild biodiversity preservation services. This management is carried out by the farmers and the government in accordance with their respective, regulated responsibilities, as economic owners of the private and public products of the *montes*, respectively.

The non-industrial individual private owners are those who manage most of the *dehesa* farm in which *Quercus* species predominate. In contrast, the public municipal owners and the regional government of Andalusia manage most of the native *Pinus* forest farm of the case-study in Andalusia. While the species of *Quercus* originate from thinning and favouring of vegetation from natural regeneration (with the exception of the afforestation event in the 1990's financed by the agricultural land set aside of cropland policy program of the European Union Common Agricultural Policy [23]. At

the time the field work was conducted for this research, the results of these one-off plantations of Mediterranean *Quercus*, compensated by the government, had not yet been inventoried. The areas of native pines have expanded thanks to historical plantations undertaken by the government and subsequent assisted regeneration of natural vegetation. The *Quercus dehesa* farm is usually located in areas of rolling plains whereas the *Pinus* farm is generally located in mountain areas at the head of watersheds, with the exception of *Pinus pinea,* which is also present on the Atlantic Ocean flat coast of the provinces of Huelva and Cádiz.

The predominance of *Quercus* in private farms is explained by the fact that, in the past, the provision of the raw materials of pasture, firewood from pruning and cork formed the basis of the open woodland economy. In contrast, public ownership of *Pinus* forests has historically been associated with plantation and other investment in long rotations, with management also aimed at the production of public interest goods and services, such as the supply of timber, mitigation of damage caused by flooding and protection against soil erosion. Although the public ownership of the pine forests has also been justified by the role of these forests in mitigating soil erosion, there is no scientific consensus on the comparative advantages of trees as opposed to shrub in the mitigation of natural erosion of soils.

The economic trends of the *dehesa* and forest farms are notably different as regards the provision of raw materials. Thus, grazing and cork predominate in the *dehesa* farm, timber and pine cones in the forest farm. However, both types of silvopastoral farms tend to coincide today with regard to the predominance among the products consumed of non-commercial services (private amenities, public recreation, landscape conservation and preservation of threatened wild biodiversity). These *dehesa* and forest farms are also comparable as regards the acceptance, by both private and public owners, non-industrial rationalities of voluntary opportunity costs of their investments in activities which generate commercial products together with other non-commercial intermediate products of amenity (ISSnca) and donation (ISSncd) services. These mainly favour the final products consumed of the private amenity and landscape conservation activities, respectively. The non-industrial private owner (this type of ownership being the most frequent in the case of *dehesa* farm), directs mixed economic management towards the rearing of controlled animals (livestock and game species), the extraction of cork and the self-consumption of private amenities. The non-industrial public owner aims management of the forest farm towards the technical and economic management of public recreational services, conservation/improvement of landscape and the preservation of threatened wild biodiversity. The public owner leases the grazing to local family-livestock keepers and hunters´ non-profit associations, thus avoiding cash losses resulting from rearing their own livestock and game using labour employees [12,13].

## 3. Concepts and Methods of the Accounting Frameworks

In this section we define the selected income and capital concepts and methods applied under the rSNA and the AAS methodologies applied in the *dehesa* and forest farms case study in Andalusia. The accounting concepts compared here have been published in authors' recent free-access articles available online [6,7,13,16,17,20]. In addition, the Supplementary Text S1 and Supplementary Tables S1–S26 present the sequence of accounts of the variables integrated in the concepts of incomes, ecosystem services and capitals valued in the case-study *dehesa* and forest farms results.

*3.1. Concepts*

3.1.1. Self-Employed Family Labour

Extensive livestock rearing in the large *dehesa* and forest farms in the centre and south-east of peninsular Spain is mainly practiced by non-industrial owners with paid employees and by part-time family owners who do not use paid labour. This partial dedication to the activity by family livestock keepers may involve investor rationales in which low or zero family labour self-compensation predominates. These rationales may even involve accepting negative net operating margins at basic prices for the livestock activity [13].

We consider that imputing market opportunity costs to the time dedicated by self-employed labour of family livestock keepers without land is an erroneous hypothesis. Thus, a consistent estimation would be to accept its residual valuation limited to a subjective hourly maximum compensation below that of paid employees for the same task. In this study up to a maximum value of 80% of the hourly remuneration for employee labour in the same activity is assumed [17], subject to the condition that the net mixed income is a positive value. Consequently, a net mixed income at negative basic prices residual value means that it all corresponds to a negative residual net operating margin at basic prices and compensation of zero for family self-employed labour.

3.1.2. Non-Commercial Intermediate Product of Services of Livestock Private Amenity Auto-Consumption

The part-time nature of the extensive livestock rearing by family keepers in the large *dehesa* and forest farms, grazing leaseholders without land or paid labour, is consistent with the payment of canons for the leasehold of grazing, zero compensation for self-employed labour and the acceptance of negative residual net operating margins at basic prices ($NOM_{bp}$) for the immobilized capital investment in livestock. These leaseholders accept the $NOM_{bp}$ in return for the enjoyment of private amenities of the extensive livestock activity.

The subjective estimation of the non-commercial intermediate product of services of private amenity auto-consumption (ISSnca) by the grazing leasehold family livestock keepers may differ in this research among the private and public *dehesa* and forest farms.

In the public farm the leaseholders tend to be owners of a small number of livestock head that enjoy the extensive livestock activity. These livestock owners accept compensations of zero value for self-employed labour and negative residual net operating margins at basic prices ($NOM_{bp}$) for the manufactured immobilized capital investment in livestock. Thus, if the leasehold non-industrial livestock keepers in public farm obtain a negative net mixed income at basic prices, they accept this opportunity cost in return for the ISSnca. The latter is estimated according to its minimum value equal to the absolute value (positive) of the net mixed income at basic prices for each species in the livestock rearing activity as a whole. If the net mixed income at basic prices is positive, then the compensation for self-employed labour is estimated first and the residual net operating margin is estimated at basic prices ($NOM_{bp}$).

In the private farm, the leasehold livestock keepers tend to be owners of a large number of livestock head that enjoy the extensive livestock activity. Herds of between 200–700 head of sheep, between 100–300 *Montanera* pigs and 80 head if they have extensive cattle. These livestock keepers can accept compensations of zero value for self-employed labour and do not forsake an assumed baseline competitive net operating margin for the immobilized capital investment in livestock. These findings are consistent with the private amenity consumer rationale of the grazing-leasehold family livestock keepers in private *dehesa* and forest farms, which we assume expect to reach an assumed baseline competitive profitability rate, per species, from the capital invested in livestock. This is the same behaviour that we have assumed in the case of *dehesa* and forest farms owners who invest in livestock rearing carried out by paid employees. Hence, in both

types of livestock ownership, we assume that the investment in the rearing of livestock, with grazing either on the livestock owners' farm or on privately-owned farm belonging to others in the case of leasehold grazing, an assumed baseline competitive operating profitability is obtained. The estimation of the ISSnca for a livestock species in the private farm requires that the assumed baseline competitive net operating margin exceed the net operating margin at basic prices, the latter being estimated after having first remunerated the self-employed labour [24].

### 3.1.3. Conservation Forestry Activity

The native Mediterranean forest species conservation activity is considered of public interest by the government of Spain and the regional government of Andalusia. Given the widespread indifference towards investment in forest restoration by the land owners, it has historically been governments that have "purchased" in advance, compensating at the costs prices of own account gross capital formation (GFCF) of forest cover restoration treatments, the "commercial" intermediate products of services (ISSc) which are used by the public activity of landscape conservation services as inputs of ordinary own compensated commercial intermediate services (SScoo). Compensating, at the costs prices of own account GFCF of forest, cover restoration treatments.

In this study, the conservation forestry activity registers all the forest vegetation treatments with the exception of those undertaken exclusively for the purpose of improving the biological productivity of grazing pasture. This concept of the conservation forestry activity justifies the fact that its total product can be made up of own commercial intermediate product of service (ISSc) and the final product of inanimate manufactured gross fixed capital formation (GFCFmi) of the plantations and constructions. The concept of the conservation forestry activity conditions the records for the timber, cork, and industrial fruit (e.g., pine nut) activities, for which the only silvicultural costs registered are those derived from the extraction of the final products harvested and the natural growth of forest vegetation at environmental prices (unit resource rent).

### 3.1.4. Forest Firefighting Activity

The public activity of forest firefighting services, which correspond to the government, aim to mitigate the degradation and destruction of woody vegetation, wild fauna, livestock grazing cultural heritage, domestic biological variety, constructions and equipment in farm landscapes. This activity differs from that of conservation forestry in that it does not include vegetation restoration/improvement treatments before and after the occurrence of forest fires. The exception, for practical reasons, is the recording of government forestry treatments in livestock ways (*cañadas*) and public trails in forest areas. The firefighting services activity records are of the same class as those of the conservation forestry activity. The ISSc of the firefighting services activity are also used as ordinary own commercial intermediate consumptions of services (SScoo) in the public landscape conservation activity.

### 3.1.5. Public Landscape Conservation Activity

Together with the livestock activity, the public landscape conservation activity is the backbone of the private and public production as a whole of the Mediterranean silvopastoral *montes* (*dehesa* and forest) farms. The production account classifies the final product into consumption (FPc) and inanimate manufactured gross fixed capital formation (GFCFmi), while the intermediate consumption is separated into purchased (ICb) and ordinary own intermediate consumption (ICoo). The latter comprises only the ordinary own intermediate consumption of services (SSoo), separated into commercial (SScoo) and non-commercial (SSncoo). The SScoo come from the conservation forestry and firefighting service activities commercial intermediate products of services (ISSc). The SSncoo come from the non-commercial intermediate products of compensations

(ISSncc) and donations (ISSncd) for farmer activities. Among the aforementioned activities, the main one are the livestock and hunting activities.

*3.2. Methods*

3.2.1. Transaction Prices under the Accounting Frameworks

The valuation of total income and capital of society from the nature-based activities of the *dehesa* and forest farms depends on the social prices of the final products consumed.

The estimation of the total sustainable income of the case-study farms requires the replacement of the rSNA non-market final product consumed valued at cost prices by exchange value stated o revealed by consumer marginal willingness to pay (MWTP). The AAS estimates these products according to their simulated exchange values at social prices. These products without market prices can be considered implicit transactions of the farmers with themselves (self-consumption of amenities) as well as between the public owners and society represented by the government (consumption of final recreational services, landscape services and threatened wild biodiversity services).

We have valued four classes of final product consumed using consumer stated preference methods (contingent valuation and choice experiment) for estimating the simulated exchange values of private amenity, public recreation, landscape and threatened wild biodiversity services. The final product consumed of water has been estimated according to its transaction prices, applying an assumed competitive real discount rate of 3% to the value of the environmental asset of water, estimated using the hedonic prices method [25]. The water transaction price coincides with the environmental price due to the absence of manufactured costs in the production function of water from the case-study farms stored further down the watershed in public reservoirs and used in irrigated croplands (see for valuation methods the Sub-Section 2.4. Forest Products Valuation, pp. 222–224 and Figure 2. Methods applied to value forest products in Andalusia, p. 223 in reference [20]).

The AAS values the final products consumed at social prices, as does the rSNA in the case of products with market prices. The rSNA values the final products without market prices consumed at cost prices. For these products, the rSNA simulates the cost prices as corresponding to the assumed social prices. This assumption implies that the products without market prices consumed generate zero value net operating margins (surpluses in SNA).

The products of a farm are registered at different stages of the production process in a given period and can be valued according to different price types until their consumption as final products when they are valued at social prices under the AAS methodology. In the case-study farms the prices vary due to the incorporation of non-commercial intermediate services of compensations (ISSncc) in the rSNA and the AAS. Additionally, in the AAS the prices are further modified due to the addition of (1) self-consumption of private amenities (ISSnca) and public owner donations (ISSncood); (2) the inputs of ordinary own non-commercial intermediate consumptions of compensation services (SSncooc), amenities (SSncooa) and donations (SSncood).

The prices of the final products consumed are estimated at producer (market) prices, basic prices and social prices. In the SNA and AAS the producer prices (pp) of a final product consumed corresponds to the market prices of the goods and services before including the operating subsidies less the products related taxes (henceforth compensations). In the rSNA, the basic prices (pb) is given by adding the government compensations (c), for the activities which receive them (ISSncc), to the value of the product at producer prices. In the AAS, besides adding the ISSncc to the producer prices, the basic prices are obtained by subtracting from the producer prices the SSncooc of the products which consume as input of intermediate consumption (the rSNA does not include the SSncooc). The commercial intermediate products of services without market prices and

the inanimate manufactured gross fixed capital formation (GFCFmi) are valued at cost prices (cp). Furthermore, the rSNA estimates the final products consumed of the private amenity and public government activities without market prices at cost prices. The AAS substitutes the rSNA cost prices of these activities for the marginal social prices (sp) of a simulated transaction stated or revealed by the users. Finally, the environmental prices (ep) correspond to the unit prices of the resource rent (it does coincide with ecosystem service) [1].

The types of prices do not change the values added of the aggregate activities of the *dehesa* and forest farms as a whole estimated under the rSNA and AAS. However, the types of prices applied do change the values added of the individual and aggregate activities of the institutional sectors of farmer and the government. In addition, the SSncoo changes the value of environmental return of environmental fixed asset.

### 3.2.2. Total Product

The total product consumed (TPc) is the consumption which takes place in the period. The final product of gross capital formation (GCF) is its expected possible consumption, as a production factor, in the value of the completed final product consumed in the future. Both types of total product (TP) for the period are valued, in some cases, based on the System of National Accounts (SNA) of observed real market transaction prices (e.g., livestock) and in others, through hypothetical markets (e.g., carbon). These simulated markets refer both to certain products consumed in the period and to other work-in-progress or finished products for future use as production factors in the same farm economic unit which produces them. The AAS methodology includes the valuations of products without market prices consumed, extending the uncertainties of the values added with respect to the valuations of the products without market prices at cost prices under the SNA methodology [2].

The total product of an activity j (TPj) is the result, in one period (year), of a technical production function f, which contains environmental and manufactured (human made) production factors. It is accepted that nature provides free environmental inputs of intermediate consumption (ICej) and services of fixed environmental assets (EFAj). Furthermore, human intervention contributes labour (Lj), manufactured intermediate consumption (ICmj) and services of manufactured fixed capital (FCmj):

$$TPj \equiv f\ (ICej,\ EFAj,\ Lj,\ ICmj,\ FCmj) \tag{1}$$

The corresponding economic equation for the f function derived from the production account expressed in monetary numeraire is Equation (2):

$$TPj = ICej + ICmj + LCj + CFCmij + CFCej + NOMmj + NOMej, \tag{2}$$

where LCj is labour compensation, CFCmj is consumption of inanimate fixed capital, CFCej is consumption of environmental fixed asset, NOMmj is manufactured net operating margin, and NOMej is environmental net operating margin.

The TPj comprises the intermediate product (IPj), the final product consumed (FPcj) and the gross capital formation (GCF). The latter made up of manufactured capital (GCFmj) and natural growth (NGcj). The total product consumed (TPcj) is obtained by aggregating the IPj and FPcj. The production factors of the TPcj are the same as those of the TPj and they are separated and referred to as 'ordinary' (o). Those production factors affecting the GCFmj and the NGj are referred to as 'investment' (i):

$$TPj = IPj + FPcj + GCFmj + NGcj \tag{3}$$

$$TPcj = IPj + FPcj \tag{4}$$

$$GCFj = GCFmj + NGcj \tag{5}$$

$$TPcj = ICeoj + ICmoj + LCoj + CFCmioj + NOMmoj + NOMeoj, \qquad (6)$$

where ICeoj is environmental work in progress used (WPeu) in this case study of *dehesa* and forest farms.

The values are known for all the components of the Equation (6) with the exception of the environmental net operating margin (NOMeo), which is estimated as the residual value of the Equation (6).

### 3.2.3. Values Added

The values added both gross (GVA) and net (NVA), represent the gross and net operating incomes which are universally used to express the importance of the individual and aggregate economic activities at national/sub-national scale. At national and regional macro spatial scales the gross value added is known as the gross domestic product (GDP). The gross value added (GVA) is also the economic variable which is most criticized in its standardized version applied in the System of National Accounts (SNA). The criticisms of the GVA relate to the omissions and biases described in the Supplementary Text S1, although its significance as operating income of the economic activities is not brought into question. This is also the point of view in this study in which the AAS methodology is applied, focusing on the concept of values added of the economic activities mitigating the biases in the SNA applications, although others are maintained which are avoided with the estimation of capital gain and total income. The capital gain is estimated based on the adjusted of capital revaluations in the balance account for the individual economic activities in each farm unit case study valued.

The net value added (NVA) provides the operating income production factor services accrue from human labour compensation (LC) and net operating margin (NOM). The latter come from the services of manufactured capital investment and environmental fixed assets appropriated by the owners of the land, the livestock and the government as trustee of the national collective ownership rights on public economic activities:

$$NVA = TP - IC - CFC \qquad (7)$$

$$NVA = LC + NOM \qquad (8)$$

$$LC = LCe + LCse \qquad (9)$$

$$NOM = NOMm + NOMe, \qquad (10)$$

where: IC is intermediate consumption, CFC is consumption of fixed capital, NOM is net operating margin, LCe is compensation of employee labour and LCse is the residual compensation of self-employed labour, in the accounting period.

The total product Equation (2) shows all the components of the AAS net value added (NVA$_{AAS}$). The criticisms of the net value added under the SNA (NVA$_{SNA}$) relate to the narrow definition of economic activity, which refers exclusively to commercial products, as well as to the inconsistent valuation of total products without market prices at cost prices rather than at simulated market prices as in the AAS.

The AAS methodology overcomes the limitations of the rSNA net value added and estimates the ordinary net value added at social prices (NVAoj$_{sp,AAS}$) of the individual total products consumed, j (TPcj) of the case-study *dehesa* and forest farms:

$$NVAoj_{sp,AAS} = TPcj - ICmoj - WPeuj - CFCmoj \qquad (11)$$

$$NVAoj_{sp,AAS} = LCoj + NOMoj \qquad (12)$$

$$NOMoj = NOMmoj + NOMeoj \qquad (13)$$

Furthermore, the AAS estimates the investment net value added ($NVAij_{sp,AAS}$) of the gross capital formation ($GCFj$). The manufactured $GCFmj$ contains the alive (livestock) $GCFmaj_{PP}$, which is valued at producer (market) prices, and the inanimate $GCFmij_{cp}$ corresponding to plantations, constructions and equipment, which is valued at cost prices. In this research, the total investment cost ($TCij$) comprises the manufactured intermediate consumptions ($ICmi$), the labour compensation ($LCij$), the manufactured fixed capital consumption ($CFCmij$) and the environmental fixed capital consumption ($CFCeij$). We distinguish between the investment net values added manufactured ($NVAmij_{PP,AAS}$) and environmental ($NVAeij_{ep,AAS}$). The inanimate manufactured investment net value added ($NVAmij_{PP,AAS}$) coincides with the labour compensation ($LCiij_{,AAS}$). The $NVAeij_{PP,AAS}$ coincides with the environmental net operating margin investment ($NOMeij$). The latter is given by the difference between the natural growth ($NGj$) and the investment environmental consumption of fixed capital ($CFCeij$). In this research the only $CFCeij$ estimated is carbon emission:

$$NVAij_{sp,AAS} = GCFj - ICmi - CFCmi - CFCei \qquad (14)$$

$$NVAij_{AAS} = NVAmij_{pp,AAS} + NVAeij_{ep,AAS} \qquad (15)$$

$$NVAmij_{pp,AAS} = GCFmj - ICmij - CFCmij \qquad (16)$$

$$GCFmiij = TCmiij \qquad (17)$$

$$NVAmiij = LCiij \qquad (18)$$

$$NVAmiaj_{pp,AAS} = GCFmaj - ICmiaj - CFCmiaj \qquad (19)$$

$$NVAeij_{ep,AAS} = NOMei_{ep} \qquad (20)$$

$$NOMeij_{ep} = NGj_{ep} - CFCeij_{ep} \qquad (21)$$

The above described value added equations, considering the total product of an individual activity, j ($TPj$), the total product consumed ($TPcj$) and the gross capital formation ($GCFj$) give the values for the ecosystem services ($ESj$) and the environmental net operating margin of investment ($NOMeij$). The latter is required to estimate the factorial distribution of the capital income ($CI$) of the individual activity, j, in its two components of manufactured capital income ($CIm$) and environmental income ($EI$) from nature (ecosystem). Furthermore, the values for the WPeu, NOMeo, NG and CFCei represent the direct links between the production and balance accounts of the *dehesa* and forest case study which allow the estimation of the environmental incomes.

### 3.2.4. Ecosystem Services

The concept of ecosystem services is a polysemic term, the use of which needs to be defined in this case given the multiple applications of the term in the literature in economics and biophysical meanings. Our objective is to assure the consistency of the sequences of interrelationships between the ecosystem services and other numbers of nature, represented in this research by the concepts of environmental assets and incomes. The recommended economic definition of ecosystem service by the United Nations Statistics Division (UNSD) should avoid the frequent polysemic problem in economic applications, which reflects the uses of the term in academic and government publications: "Ecosystem services (ES) are the contributions of ecosystems to benefits used [consump-

tion of economic products] in economic and other human activity" [1]. We think it is more constructive to explain our own method of calculation of the ecosystem service which we believe can also fit into the definition of the concept recommended by the reference [1].

The concept of ecosystem service in this research refers to the free contribution of nature (ecosystem) to the economic value of a total product consumed, j, at social prices (TPcj) of the case-study *dehesa* and forest farms. The Equation (22) shows that the possible contribution of the ecosystem comes from the ICeoj, CFCeoj and NOMeoj under the AAS methodology:

$$ESj = ICeoj + CFCeoj + NOMeoj, \qquad (22)$$

where CFCeoj is ordinary environmental consumption of fixed capital.

In this research we have not registered the degradation of the ordinary environmental consumption of fixed capital (CFCeoj) among the production factors of the TPc of the case-study *dehesa* and forest farms, although it is considered in the estimation of the environmental asset revaluation (EAr). The only ICeo are the environmental work-in-progress used (extracted) of timber, firewood and cork (WPeu) valued according to their environmental prices at the opening of the period. Thus, in this research we register the components of the ecosystem services of the *dehesa* and forest farms shown in the Equation (23):

$$ESj = WPeuj + NOMeoj \qquad (23)$$

The coexistence of the two components of the ecosystem service in the same product is due to accounting conventions and the origin of the products. For example, in this research the ecosystem services (ES) of the captures of game species inventoried at the opening of the period are considered WPeuj, while the captures of non-inventoried and migrant game species may be considered as NOMeoj.

The ES are valued at environmental prices and in the case of farmer commercial activities are not affected by the incorporation of non-commercial intermediate products of compensation (ISSncc), private amenity (ISSnca) and donation (ISSncc) services. However, the counterparts of the ISSncc/a/d as inputs of own ordinary non-commercial intermediate consumptions of services (SSncooc/a/d) affect the values of the ecosystem services of activities for which the final products consumed benefits. In this research the activities mainly affected by the SSncooc/a/d are the non-commercial activities of private amenity and landscape conservation services.

### 3.2.5. Environmental Asset

The total capital value (C) at the close of the period depends on future events which we assume to be replicated through human intervention for an infinite time horizon. The level of uncertainty in the valuation of the different capitals will vary depending on the product, although even the products for which there is a direct market price (e.g., livestock, machinery, buildings) may be subject to extraordinary destruction. An additional uncertainty for biological assets is their dependence on the variability of environmental conditions under a Mediterranean climate amongst seasons and years changes. In this research we assume that the anticipated future events remain the same as in the current period, except in the case of woody products and fruit (acorn and pine nuts), which depend on the biological cycles of the existing inventoried trees and their successive replacements. In each period, the discrepancies between the opening and closing capital values are embedded in the capital gain (CG).

An important factor affecting the degree of uncertainty in the valuation of environmental assets of the *dehesa* and forest farms is the subjective choice of the assumed competitive real discount rate. Another factor in this respect is the assumption in this research concerning the real availability for possible sale of the case-study *dehesa* and

forest farms belonging to non-industrial public owners [26]. If it were legally possible for a public farm to be sold, it is highly unlikely in practice that this future sale would occur within a time horizon which would have a significant effect on the value of the environmental asset. The effect of the hypothetical removal of public farm from the land market on the valuation of the private amenity environmental asset is discussed below.

The AAS separates the total capital (Cj) of the individual activity, j, into manufactured capital (Cmj) and environmental asset (EAj). The AAS classification of the capital (Cj) of individual activities also distinguishes between the capital of work-in-progress used (WPj) and fixed capital (FCj). The latter, in turn, is separated into land (FClj), biological resources (FCbrj) and manufactured capital (FCmj). There is no conceptual controversy as regards the valuation of manufactured capital and we focus on the valuation of the environmental assets:

$$Cj = WPj + FCj \qquad (24)$$

$$WPj = WPmj + WPej \qquad (25)$$

$$FCj = FClj + FCbrj + FCmj \qquad (26)$$

$$EAj = WPej + FClj + FCbrj \qquad (27)$$

The value of an environmental asset is based, on the one hand, on the present discounted value of the flows of future economic resource rent up to the final period in which it is exhausted or for an infinite time horizon if the environmental asset is not consumable, and on the other, on factors autonomous from the products consumed (A). In this research, the economic rent from products consumed in the period is termed natural resource rent (RR). In other words, the RR and the ES coincide with environmental income in the farm in which the autonomous factor (A) and the depletion (WPeu) adjusted change of environmental net worth (CNWead) is zero.

The autonomous factor of the environmental asset stems from its non-reproducible fixed component of land, whereby the land market can act as a 'store' for value, fiscal mitigation and other factors unrelated to the output of nature-based product active and passive uses. In this research we omit the autonomous component of the environmental asset in the case of private owners of *dehesa* and forest farms given the small contribution of the autonomous component to the value of the environmental assets (see Table 8, p. 45; Table A.4.14, p. 129, in reference [22]).

### 3.2.6. Total Income

The total income (TI) of an individual activity, j, and of the activities of the *dehesa* and forest farms as a whole corresponds to the maximum possible consumption of the total product while maintaining the same real capital value at the close of the period as at the beginning of the period [3] (p. 85). In the AAS methodology the value added corresponds to the operating income. This represents the estimation of the part of the farm income originating in the total product. The value added estimated by the AAS is an operating income with a bias resulting from the overvaluation of the natural growth of woody products and the final product consumed of carbon. These overvaluations are due to both the natural growth and carbon final product being counted in their opening environmental assets, so the real value added should only be the revaluations in the period. Furthermore, the value added does not take into account the fact that the economic activities with environmental work-in-progress inventoried at the beginning of the period and still present at the close of the period generate a real revaluation due to the reduction, by one period, in the time remaining until the future harvest of the product. It is necessary to incorporate the capital gain (CG) in order to estimate the sustainable total income of the *monte* (see Equation A.3, p. 192, in reference [27]). Thus, for an individual activity j:

$$TI = NVA + CG \tag{28}$$

The AAS capital balance account registers the revaluations (Cr), destructions (Cd) and instrumental adjustments (Cad) of the manufactured fixed capital consumptions, the natural growth of woody products inventoried at the close of the period and the final product consumed of carbon. These records make it possible to estimate the capital gain (GC). In the AAS methodology the capital gain is separated into manufactured (GCm) and environmental (EAg):

$$CG = Cr - Cd - Cad \tag{29}$$

$$CG = CGm + EAg \tag{30}$$

The EAg is estimated according to the revaluation (EAr) less instrumental adjustment to environmental asset (EAad) for avoiding total income double counting. Being the EAad the natural growth of wood products inventoried in the period close (NG), the final product consumed of carbon (FPcca) valued at the beginning of the period, and other adjustment (EAado).

$$EAg = EAr - EAad \tag{31}$$

$$EAad = NG/(1 + r) + FPcca/(1 + r) + EAado \tag{32}$$

The AAS production and capital balance account records allow the total incomes to be estimated. They also allow the factorial distributions to be obtained for the individual activities of the case-study *dehesa* and forest farms. We separate the distribution of the production factors into labour compensation (LC), manufactured capital income (CIm) and environmental income (EI):

$$TI = LC + CIm + EI \tag{33}$$

The environmental income (EI) is estimated by aggregating the environmental net operating margin (NOMe) and the environmental asset gain (EAg). The direct reorganization of these components gives another equation for the EI as the aggregate of the ordinary environmental net operating margin (NOMeo) and the change in environmental net worth (CNWe):

$$EI = NOMe + EAg \tag{34}$$

$$EI = NOMeo + CNWe \tag{35}$$

$$CNWe = NGc - CFCei + EAg \tag{36}$$

We aim to present the environmental income composed of the ecosystem service (ES) and the depletion (WPeu) adjusted change of environmental net worth (CNWead):

$$EI = ES + CNWead \tag{37}$$

$$CNWead = CNWe - WPeu \tag{38}$$

### 3.2.7. Social Profitability Rates

The results for the profitability rates are conditioned by the hypothesis in this research that the total products consumed of the individual activities reach at least an assumed competitive 3% real profitability rates for ordinary manufactured immobilized capital (IMCmo), except in the case of residential and commercial recreation service ac-



tivities, for which the residual net operating margins are estimated at producer (market) prices. As regards the activities for which the hypothesis of obtaining an assumed baseline competitive ordinary manufactured net operating margin (NOMmoc) is applicable, except for livestock which is assumed a competitive total net operating margin. We incorporate the non-commercial opportunity cost incurred voluntarily by the farmer (VOC) in the intermediate product and in the non-commercial intermediate consumption of the corresponding individual economic activities. This VOC corresponds to the non-commercial intermediate product of services of private amenity self-consumption (ISSnca) and donation (ISSncd).

The ordinary profitability rates (Po) of the individual activities under the AAS include the private amenities (ISSnca) and public owner donations (ISSncd) along with their counterparts of intermediate consumption of private amenities (SSncooa) and of donations (SSncood). Since the SSncood are registered in the intermediate consumption of the public activities, it not only affects the profitability of the individual activity which generates the ISSncd, but also the aggregate profitability of the farmer activities. The inclusion of the ISSnca and SSncooa only affects the individual activities but not the aggregate activities of the farmer. The net operating margin overvaluation biases imply additional uncertainty factors, thus the need to interpret with caution the robustness of the subjective estimates of the ordinary profitability rates. These biases are due to the inclusion of carbon fixation and natural growth of woody products in the final product according to their market prices at the closing of the period. In the AAS these two final products are valued in the opening asset and are not taken into account as inputs of their respective products in the period. The level of uncertainty with regard to the profitability rates of the capital gain (Pg) in a period is greater than that for the Po due to the volatility of the prices of inanimate manufactured fixed capital and of land. Given these weaknesses of the two rates which comprise the total profitability rate of an individual activity and of the aggregate activities of the case-study *dehesa* and forest farms as a whole, it was decided that they should be presented separately.

The operating (Po) and capital gain (Pg) profitability rates are estimated according to the ratios between the net operating margin (NOM) and the capital gain (CG) for the immobilized capital (IMC) in the period. The total profitability rate (P) of the capital income (CI) of the economic activity is estimated by adding both profitability rates at social prices:

$$IMC = Co + 0.5 \times (Cb + TC - RMo - WPu - CFC - FPs - Cs) \tag{39}$$

$$Po = NOM/IMC \tag{40}$$

$$Pg = CG/IMC \tag{41}$$

$$P = CI/IMC, \tag{42}$$

where Co is the opening capital, Cb is bought (purchased) capital, RMo is own raw material consumed in the production process, WPu is work in progress used; FPs are the final products sold and Cs are capital sales arising during the period. The total cost (TC) comprises intermediate consumption (IC), labour (LC) and consumption of fixed capital (CFC) (see details in reference [13], SM S1, p. 3, SE9).

The rates of profitability (return) of wood and non-wood forest products at social prices estimated by the AAS include the manufactured net operating margins (profits) of the non-commercial intermediate product of services (ISSnc) and their counterparts of own ordinary non- commercial intermediate consumption of services (SSncoo). In addition to the abovementioned extension of rSNA, the AAS incorporates wood and non-wood environmental natural growth in total products, that is, of wood and game products less investment consumption of environmental fixed asset of carbon released.

The rSNA does not incorporate NG or carbon activity in the measurement of net operating surplus at basic prices. The farmer total profitability rate at social prices in the rSNA does not include the private amenity environmental net operating surplus. Meanwhile, the private amenity EFA in the rSNA is included in the measurement of total land market price. Due to these limitations of the standard SNA it is not consistent to compare the total profitability rates of the farmer activities measured by the AAS and SNA.

*3.3. The Refined Standard System of National Accounts Applied in Dehesa and Forest Farms Case Study*

In this research, the slight refinement of the standard System of National Accounts (rSNA) consists of registering the commercial intermediate product of grazing, separating the net mixed income (NMI) of the individual activity into imputed compensation of unpaid (self-employed) labour and net operating surplus, subtracting the environmental work-in-progress used from the net operating surplus and incorporating it in the intermediate consumption of the individual activity. In addition, the government compensation (operating subsides net of taxes on production) is incorporated as non-commercial intermediate product of service (ISSncc), and their counterpart of ordinary own non-commercial intermediate consumption of service (SSncoo). These changes are intended to make the intermediate product, intermediate consumption and labour compensation under the rSNA and AAS consistently comparable.

The rSNA estimates the total capital of the farmer at market prices, implicitly including the environmental fixed assets of the private amenity in the land market value. The government activities valued by the rSNA include the manufactured capital at market prices of fire services, recreation services, landscape conservation services and threatened wild biodiversity preservation services. The only government activities for which the rSNA estimates the environmental fixed asset values are those of mushrooms and water runoff (water yield) with economic use in irrigated land further down the watershed. The rSNA omits the carbon activity in the *dehesa* and forest farms case study (see Supplementary Text S1).

## 4. Results

We present the same economic indicators for the four groups into which we have classified the 41 case-study *dehesa* and forest farms. These groups comprise 21 private and five public *Quercus* open woodland *dehesa* farms, along with 13 public and two private *Pinus* forest farms. We prioritize the presentation of the results for the incomes, ecosystem services and profitability rates for the individual activities, those of the farmer, the government and the farms as a whole. The results for total incomes and environmental income include their respective net values added, capital gains and net environmental margins, respectively.

The case-study *Quercus dehesa* and *Pinus* forest farms illustrate the trends which they share with their respective genera and ownership types. Due to the small number of public *dehesa* and private forest farms in the case study, there is a greater degree of uncertainty as regards the robustness of the results in comparison to those for the private *dehesa* and public forest farms, of which there are a much larger number.

We do not present aggregate physical and economic indicator estimates for the 41 case-study *dehesa* and forest farms as a whole. We consider that the absolute aggregate results for these case-study farms may not be representative of the possible mean results, weighted by area, for the four groups of farms classified according to ownership types and genera of the trees. Hence, the percentage area per tree species and ownership types does not correspond to their respective percentages at the regional scale of Andalusia.

We first present a summary of grazing consumption, followed by the most noteworthy economic results for the four groups of silvopastoral farms under the AAS. We compare the AAS and rSNA results for the gross values added and ecosystem services in

the two case-study farm groups of private *dehesa* and public forest farms. Finally, we compare the sensitivity of GVA and ES to prices types under the AAS.

### 4.1. Livestock Stocking Rate and Grazing Consumption of Forage Units

#### 4.1.1. Livestock Stocking Rate

The vegetation in the private and public *dehesas* mainly consists of trees of the *Quercus* genus along with scrub, pasture and cultivated land, all of which is grazed by controlled livestock and game species during all or part of the accounting period (year) in the case study (Table 1). The inventories of livestock that graze the *dehesa* farms are: 44 LU/km$^2$ in the private *dehesa* and 2.4 LU/km$^2$ in the public *dehesa* (Table S1).

In the predominantly public and private conifer forest farms, grazing of domestic and game animals mainly occurs in areas not covered by the *Pinus* species. Therefore, in these predominantly conifer forest farms it is also the areas of *Quercus* woodland, shrub, pasture and cultivated cropland where most of the grazing occurs. The inventories of livestock that graze the forest farms are: 5.9 LU/km$^2$ in the private farm and 2.0 LU/km$^2$ in the public farm (Table S1).

#### 4.1.2. Livestock and Game Species Grazing Forage Unit Consumption

In 2010, the case-study results reveal that consumption of grazing by livestock and game species accounted for 50.4% and 94.5%, respectively, of their total forage unit consumption. Grazing makes up 57.6% and 92.5%, respectively, of the food of inventoried domestic livestock and game species present in the private and public *dehesas* (Tables 2, S2 and S3). Game species consume 49.0% of the grazing and acorns in the private *dehesa* and 79.1% in the public *dehesa* (Tables 2, S2 and S3). However, grazing consumption in the private *dehesa* is 3.2 times that of the public *dehesa* since the area of *Quercus* is notably lower in the latter (Tables 2, S2 and S3). The private *dehesa* of *Quercus* (PR, OD) is agroforestry farm with considerable consumption of grazing fodder by livestock and a much lower intensity of consumption by game species.

**Table 2.** Grazed fodder consumption of case-study large *dehesa* and forest farms in Andalusia (2010: FU/ha *).

| Class | Privately-Owned *dehesa* Farm | | | Publicly-Owned *dehesa* Farm | | | Privately-Owned Forest Farm | | | Publicly-Owned Forest Farm | | |
|---|---|---|---|---|---|---|---|---|---|---|---|---|
| | Grazing | Supplements | Total | Grazing | Supplements | Total | Grazing | Supplements | Total | Grazing | Supplements | Total |
| Grazing livestock | 288.5 | 386.6 | 675.1 | 36.8 | 10.6 | 47.5 | 20.5 | 118.8 | 139.3 | 42.9 | 19.9 | 62.9 |
| Hunting | 277.2 | 29.1 | 306.3 | 139.6 | 3.7 | 143.3 | 208.2 | 8.6 | 216.8 | 44.7 | 0.4 | 45.1 |
| Total | 565.7 | 415.7 | 981.4 | 176.5 | 14.3 | 190.8 | 228.7 | 127.4 | 356.1 | 87.6 | 20.3 | 107.9 |

* FU: A forage unit is the energy content of a kilogram of barley with a humidity content of 14.1% and totals 2723 kcal [28].

In the public forest farm (PU, PF) there is scarce grazing consumption by controlled animals, with a predominance of game species grazing. Game species consume 51.0% of the total forage units grazed in the public forest farm and 91.0% in the private forest farm (Tables 2, S4 and S5). The grazed forage unit consumption (FUg) of livestock and game species in the private *dehesa* farm (981.4 FUg$_{PR, OD}$/ha) is 9.1 times greater than the FUg of the public forest farm (107.9 FUg$_{PU, PF}$/ha) (Tables 2, S2 and S5). The livestock forage unit consumption per unit area (FUg) in the private *dehesa* farm (675.1 FUgli$_{PR, OD}$/ha) is 10.7 times greater than that of the public forest farm (62.9 FUgli$_{PU, PF}$/ha) (Tables 2, S2 and S5). Game species consume 6.8 times more FUg in the private *dehesa* farm (306.3 FUghu$_{PR, OD}$/ha) than in the public forest farm (45.1 FUghu$_{PU, PF}$/ha) (Tables 2, S2 and S5).

### 4.2. Net Values Added

The case study total products (TP) of the *dehesa* and forest farms have intermediate products (IP) embedded in the final product consumed (FPc). This double counting does not affect the estimates of farm values added as the double counting of the intermediate

product is neutralised by the recording of the ordinary own intermediate consumption (Table 3). However, measurements of the individual activities intermediate products and ordinary own intermediate consumption are required to estimate their ordinary net operating margins assumed to be competitive as a lower bound. In addition, there are activities whose intermediate products constitute the main contribution of the activity to the total product (e.g., grazing, residential service, conservation forestry service, fire service).

**Table 3.** Production and income generation account of case-study large *dehesa* and forest farms in Andalusia under the AAS (2010: €/ha).

| Class | Private-ly-Owned *dehesa* Farm | Public-ly-Owned *dehesa* Farm | Private-ly-Owned Forest Farm | Public-ly-Owned Forest Farm |
|---|---|---|---|---|
| 1. Total product at social prices (TP$_{sp}$) | 1474.2 | 775.5 | 812.0 | 487.7 |
| 1.1 Total product consumption at social prices (TPc$_{sp}$) | 1211.5 | 734.4 | 744.0 | 468.9 |
| 1.1.1 Intermediate product at social prices (IP$_{sp}$) | 348.2 | 144.6 | 192.4 | 85.2 |
| 1.1.1.1 Intermediate raw material at producer prices (IRM$_{pp}$) | 82.4 | 29.2 | 17.6 | 6.7 |
| 1.1.1.2 Intermediate services at producer and social prices (ISS$_{pp,sp}$) | 265.7 | 115.4 | 174.9 | 78.5 |
| 1.1.2 Final product consumption at social prices (FPc$_{sp}$) | 863.3 | 589.8 | 551.5 | 383.7 |
| 1.1.2.1 Market product | 251.1 | 51.8 | 41.3 | 22.2 |
| Timber | | 13.9 | 0.5 | 9.1 |
| Cork | 63.2 | 22.6 | | |
| Firewood | 5.8 | 0.1 | 0.2 | 0.0 |
| Nuts | | 0.0 | 12.4 | 0.8 |
| Grazing | 0.4 | 0.1 | | 0.2 |
| Aromatic plants | | | | 1.9 |
| Hunting | 22.3 | 9.4 | 2.1 | 5.4 |
| Commercial recreation | 0.3 | | | |
| Residential | 4.2 | 0.4 | 3.0 | 0.3 |
| Livestock | 133.0 | 5.2 | 23.2 | 4.2 |
| Agricultural crops | 21.9 | | | 0.1 |
| 1.1.2.2 Non-market product | 612.2 | 538.0 | 510.2 | 361.5 |
| Amenity services | 311.2 | 1.6 | 356.8 | 2.5 |
| Recreation services | 26.2 | 99.3 | 9.9 | 80.4 |
| Mushrooms | 13.4 | 15.5 | 13.5 | 5.7 |
| Carbon | 52.3 | 85.3 | 56.9 | 57.5 |
| Landscape | 126.4 | 196.0 | 55.1 | 143.7 |
| Biodiversity | 16.7 | 30.3 | 5.3 | 22.8 |
| Water supply runoff | 65.9 | 109.9 | 12.8 | 48.9 |
| 1.2 Gross capital formation (GCF) | 262.7 | 41.1 | 68.0 | 18.8 |
| 1.2.1 Manufactured gross capital formation (GCFm) | 226.5 | 15.0 | 59.7 | 10.5 |
| 1.2.2 Natural growth at environmental prices (NG$_{ep}$) | 36.2 | 26.1 | 8.3 | 8.3 |
| Timber | 0.1 | 0.8 | 0.1 | 5.8 |
| Cork | 24.9 | 19.8 | | |
| Firewood | 0.3 | 0.1 | 0.0 | 0.0 |
| Hunting | 10.9 | 5.3 | 8.1 | 2.5 |
| 2. Intermediate consumption (IC) | 799.0 | 244.7 | 326.3 | 137.0 |
| 2.1 Manufactured intermediate consumption (ICm) | 554.6 | 215.1 | 281.2 | 132.6 |
| 2.1.1 Bought (ICmb) | 206.4 | 70.5 | 88.8 | 47.4 |
| 2.1.2 Own (ICmo) | 348.2 | 144.6 | 192.4 | 85.2 |
| 2.2 Work in progress used (WPu) | 244.4 | 29.6 | 45.1 | 4.4 |
| 2.2.1 Manufactured (WPmu) | 190.4 | 3.9 | 44.3 | 1.0 |
| 2.2.2 Environmental (WPeu) | 53.9 | 25.7 | 0.8 | 3.4 |
| 3. Gross value added at social prices (GVA$_{sp}$) (1–2) | 675.2 | 530.8 | 485.7 | 350.7 |
| 4. Consumption of fixed capital (CFC$_{rp,ep}$) | 66.4 | 40.0 | 58.1 | 33.2 |

| Class | Private-ly-Owned *dehesa* Farm | Public-ly-Owned *dehesa* Farm | Private-ly-Owned Forest Farm | Public-ly-Owned Forest Farm |
|---|---|---|---|---|
| 4.1 Manufactured ($CFCm_{rp}$) | 51.5 | 9.8 | 43.8 | 7.8 |
| 4.2 Environmental ($CFCe_{ep}$) | 14.9 | 30.2 | 14.3 | 25.4 |
| 5. Net value added at social prices ($NVA_{sp}$) (3–4 = 5.1 + 5.2) | 608.8 | 490.8 | 427.6 | 317.5 |
| 5.1 Labour compensation (LC) | 146.6 | 85.2 | 77.3 | 71.0 |
| 5.1.1 Employee | 144.8 | 84.1 | 77.3 | 70.3 |
| 5.1.2 Self-employed | 1.8 | 1.1 | | 0.7 |
| 5.2 Net operating margin a social prices ($NOM_{sp}$) | 462.2 | 405.6 | 350.2 | 246.5 |
| 5.2.1 Manufactured net operating margin a social prices ($NOMm_{sp}$) | 86.8 | 15.0 | 21.1 | 8.3 |
| 5.2.1.1 Ordinary ($NOMmo_{sp}$) | −32.9 | 11.3 | 1.0 | 7.5 |
| 5.2.1.2 Investment ($NOMmi_{pp}$) | 119.8 | 3.6 | 20.2 | 0.8 |
| 5.2.2 Environmental net operating margin a social prices ($NOMe_{sp}$) | 375.4 | 390.6 | 329.1 | 238.2 |
| 5.2.2.1 Ordinary ($NOMeo_{sp}$) | 354.0 | 394.7 | 335.1 | 255.3 |
| 5.2.2.2 Investment ($NOMei_{sp}$) | 21.4 | −4.1 | −6.0 | −17.1 |

Total product (TP) incorporates the sum of all the factors of production employed in its generation. The sequence of the process of the creation of total product starts with intermediate consumption (IC) which is manufactured through the contributions of the services of human labour (LC) and the user cost of fixed capital. The latter are composed of the consumption of fixed capital (CFC) and net operating margin (NOM). The change in the value of intermediate consumption constitutes the gross value added (GVA) embedded in total product (Table 3). The operating income is represented by the net value added (NVA) which is estimated by subtracting the consumption of fixed capital from the GVA (Table 3).

The production and generation of income account has among its factors of production of the total products of the farms the components of the resource rents provided free by nature (ecosystem services). These components of ecosystem services are represented in the case study of *dehesa* and forest farms by the environmental works in progress used (WPeu) and the environmental net operating margins (NOMeo) (Table 3).

Farms grouped by vegetation of oak woodlands *dehesa* and conifer forest farms, and private and public landowner types offer contrastingly different results (Table 3). However, the differences between the vegetation types are influenced by the locations of mostly oak woodland *dehesa* farms in the sierra areas of eastern Andalusia (Figure 1).

From the application of the production and income generation account of society represented by the AAS methodology, the perceptions of policy makers and academic experts represented in the quotes [26,29] are confirmed in the results derived from the application of the AAS methodology in the four groups of *dehesa* and forest farms in the Andalusia case study (Table 3). Thus, the predominance of environmental services among the final products consumed from the forests is indisputable. This is also true for the contributions of environmental net operating margins in all farm types (Table 3). However, the results of this account of society is the result of the economic agents' behaviour represented by the owners and governments independently taking the risks of manufactured capital investments in the productive management of the economic activities of the case study *dehesa* and forest farms. It is from this perspective of the economic rationality of the institutional sectors of owners and governments that we are interested in describing the economic results of the case study of the Andalusian *dehesa* and forest farms.

In all the case-study *dehesa* and forest farms measured under the AAS, market final products consumed are minor products ranging from 6% lower bound to 29% upper bound of final product consumption in publicly-owned forest farms and private-ly-owned *dehesa* farms, respectively (Table 3). The net value added of wood products of

the forest only contributes 1.2% lower bound to 8.6% upper bound to the net value added of the farm in the privately-owned forest farm and privately-owned *dehesa* farm, respectively (Table 4).

### 4.2.1. *Dehesa* Farm

In the large privately-owned *dehesa* of the case-study, cork contributes 45.8 €/ha to the farmer net value added at social prices of 377.9 €/ha (Tables 4 and S6). The net value added of the activities of grazing, livestock rearing and hunting is de 188.3 €/ha, which is only 1.6 times greater than the 119.1 €/ha corresponding to the private amenity activity (Tables 4 and S6). The farmer net value added is 1.6 times that of the government activities (230.9 €/ha). The net value added corresponding to the government makes up 37.9% of the total net value added of 608.8 €/ha for the private *dehesa* (Tables 4 and S6). The net values added of the government activities, from largest to smallest contributions, are those of the water, landscape conservation, carbon, firefighting services and recreational activities. However, mushroom picking and the threatened wild biodiversity preservation services are also of importance in the private *dehesa* (Tables 4 and S6).

**Table 4.** Net value added of case-study large *dehesa* and forest farms in Andalusia under the AAS (2010: €/ha).

| Class | Privately-Owned *dehesa* farm | | | Publicly-Owned *dehesa* Farm | | | Privately-Owned Forest Farm | | | Publicly-Owned Forest Farm | | |
|---|---|---|---|---|---|---|---|---|---|---|---|---|
| | $NOM_{sp}$ | LC | $NVA_{sp}$ | $NOM_{sp}$ | LC | $NVA_{sp}$ | $NOM_{sp}$ | LC | $NVA_{sp}$ | $NOM_{sp}$ | LC | $NVA_{sp}$ |
| 1. Farmer | 271.3 | 106.5 | 377.9 | 45.9 | 30.7 | 76.6 | 275.1 | 48.6 | 323.8 | 20.0 | 21.8 | 41.7 |
| 1.1 Wood forest products | 33.5 | 18.6 | 52.1 | 24.9 | 10.2 | 35.1 | 0.3 | 4.9 | 5.2 | 6.4 | 4.2 | 10.6 |
| Timber | 0.7 | 0.4 | 1.1 | 4.3 | 6.0 | 10.3 | 0.3 | 4.6 | 5.0 | 6.4 | 4.2 | 10.5 |
| Cork | 29.7 | 16.1 | 45.8 | 20.5 | 3.9 | 24.4 | | | | | | |
| Firewood | 3.1 | 2.1 | 5.2 | 0.1 | 0.3 | 0.4 | 0.0 | 0.2 | 0.2 | 0.0 | | 0.0 |
| 1.2 Non-wood forest products | 237.8 | 88.0 | 325.8 | 21.1 | 20.4 | 41.5 | 274.8 | 43.8 | 318.6 | 13.6 | 17.6 | 31.2 |
| Nuts | | | | 0.1 | 3.4 | 3.5 | 4.0 | 3.8 | 7.7 | 0.2 | 0.4 | 0.6 |
| Grazing | 52.1 | 14.1 | 66.2 | 1.5 | 0.0 | 1.6 | 13.4 | 1.2 | 14.6 | 1.2 | | 1.2 |
| Conservation forestry | 0.2 | 0.7 | 0.9 | 0.3 | 5.1 | 5.4 | 0.7 | | 0.7 | 1.2 | 11.7 | 12.9 |
| Aromatic plants | | | | | | | | | | 0.5 | 1.5 | 2.0 |
| Hunting | 32.0 | 14.5 | 46.5 | 15.3 | 10.4 | 25.7 | 11.0 | 11.9 | 23.0 | 7.6 | 3.3 | 10.9 |
| Commercial recreation | −0.7 | 0.0 | −0.6 | 0.0 | 0.0 | 0.0 | | | | | | |
| Residential | −3.5 | 6.9 | 3.4 | 0.4 | | 0.4 | 1.5 | | 1.5 | 0.3 | | 0.3 |
| Livestock | 30.3 | 45.2 | 75.6 | 3.4 | 1.6 | 4.9 | 9.5 | 21.8 | 31.3 | 2.5 | 0.7 | 3.2 |
| Agricultural crops | 8.2 | 6.4 | 14.6 | 0.0 | 0.0 | 0.0 | 4.0 | 5.1 | 9.1 | 0.1 | 0.0 | 0.1 |
| Amenity services | 119.1 | | 119.1 | | | | 230.6 | | 230.6 | | | |
| 2. Government non-wood forest products | 190.9 | 40.1 | 230.9 | 359.7 | 54.5 | 414.2 | 75.1 | 28.7 | 103.8 | 226.6 | 49.2 | 275.8 |
| Fire services | | 25.1 | 25.1 | | 28.6 | 28.6 | | 19.2 | 19.2 | | 25.7 | 25.7 |
| Recreation services | 18.4 | 4.2 | 22.6 | 87.9 | 7.1 | 95.0 | 5.2 | 2.0 | 7.2 | 68.1 | 8.9 | 77.0 |
| Mushrooms | 13.2 | 0.1 | 13.4 | 15.3 | 0.2 | 15.5 | 13.2 | 0.2 | 13.4 | 5.6 | 0.1 | 5.6 |
| Carbon | 37.5 | | 37.5 | 55.1 | | 55.1 | 42.7 | | 42.7 | 32.1 | | 32.1 |
| Landscape | 44.3 | 6.7 | 51.0 | 72.4 | 13.4 | 85.9 | 0.0 | 4.0 | 4.0 | 55.1 | 10.2 | 65.3 |
| Biodiversity | 11.6 | 3.9 | 15.5 | 19.0 | 5.2 | 24.2 | 1.3 | 3.3 | 4.5 | 16.8 | 4.3 | 21.1 |
| Water supply runoff | 65.9 | | 65.9 | 109.9 | | 109.9 | 12.8 | | 12.8 | 48.9 | | 48.9 |
| Farm (1 + 2) | 462.2 | 146.6 | 608.8 | 405.6 | 85.2 | 490.8 | 350.2 | 77.3 | 427.6 | 246.5 | 71.0 | 317.5 |

Abbreviations: $NOM_{sp}$ is net operating margin at social prices; LC is labour compensation; $NVA_{sp}$ is net value added at social prices.

In the private *dehesa* the net operating margin makes up 76.3% of the total net value added. Labour compensation account for 28.2%, 17.3% and 24.1% of the net values added of the farmer, government and private *dehesa* farm as a whole, respectively (Tables 4 and S6). Cork, livestock and hunting make up 15.1%, 42.5% and 13.6%, respectively, of the labour cost of the farmer. The farmer employs most of the labour, the compensation of which is 2.7 times greater than the government compensation for labour (Tables 4 and S6). Employee labour compensation makes up 98.3% of the total labour compensation.

The net values added at social prices for the owners of large public *dehesa* is only 20.3% of the average for those of private *dehesa* (Tables 4, S6 and S7). This is due to the lower contribution of cork, the very low value for consumed grazing products as a result of the scarce livestock herds which graze in the public *dehesa*, and particularly because of the absence of private amenity net valued added (Tables 4 and S7). In contrast, the net values added for the public activities managed either directly by the government or through contracted companies are 1.8 times greater than the respective average net values added per hectare of the private *dehesa* (Tables 4, S6 and S7). This latter result is due to environmental conditions of greater water production and consumption of landscape conservation, public recreation and carbon activities (Tables 4 and S7). Despite the lower net values added of the farmer and the higher government net values added, the total net value added per hectare in the public *dehesa* is 80.6% that of the private *dehesa* (Tables 4, S6 and S7).

### 4.2.2. Forest Farm

In the case-study large publicly-owned forest farm the timber, hunting and conservation forestry activities generate similar net values added, making up 82.2% of the net values added of the farmer at social prices of 41.7 €/ha (Tables 4 and S8). The net values added of the grazing consumed by livestock are almost non-existent. However, the ecosystem service of game species in the case-study forest farm may be similar to the value for grazing consumption by these species in the period. Thus, the "virtual" net value added of the grazing consumed by controlled animals (livestock and game) amounts to 6.8 €/ha (Table S8). The net value added at social prices for government activities in the public forest is 6.6 times greater than that of the farmers. The government activities, with a net value added of 275.8 €/ha, account for 86.9% of the total net value added at social prices of 317.5 €/ha, on average, for the public forest (Tables 3 and S8). The net values added for public activities from largest to smallest amounts are: public recreation services, landscape conservation services, water, carbon, firefighting services, threatened wild biodiversity preserservation services and mushrooms (Tables 4 and S8).

In the public forest, the net operating margin accounts for 77.6% of the total net value added of the farm. Labour compensations of the farmer, government and farm as a whole make up 52.2%, 17.8% and 22.4%, respectively, of the net values added (Tables 4 and S8). The conservation forestry, timber and hunting activities make up 53.8%, 19.1% and 15.0%, respectively, of the labour compensation of the farmer, which is 21.8 €/ha. The labour compensation of the government activities is 2.3 times that of the farmer activities (Tables 4 and S8). Employee labour accounts for 97.0% of the total labour compensation of the public forest.

The case-study private forest farm generates net values added for the farmer which are 3.1 times those corresponding to the government (Tables 4 and S9). This is due to the notably higher contributions to the net values added of livestock farming, hunting, grazing, pine cones and agricultural crops among the commercial products and particularly, due to the private amenity activity (Tables 4 and S9). The environmental conditions in the private forest farm mean lower production of water, landscape conservation, public recreation and threatened wild biodiversity preservation activities (Tables 4 and S9). The private forest farm generates a total net value added per hectare which is 1.3 times that of the public forest farm (Tables 4, S8 and S9).

### *4.3. Ecosystem Service*

Ecosystem services (ES) of wood forest products contribute with only small 0.5 €/ha and 1.9 €/ha to the final product consumed from publicly and privately owned forest farms, respectively. While the ES of wood forest products contribute to 3.6% and 5.0% to the final products consumed from publicly and privately owned *dehesa* farms, respectively (Tables 3 and 5). In relative terms, the ecosystem services of wood forest products account for 10.6% and 5.0% of the ecosystem services of private and public *dehesa* farms,

respectively (Table 5). In these farms, it is cork harvesting that explains the main contributions of ecosystem services of wood forest products to the final products consumed and ecosystem services of all activities on the *dehesa* farms.

### 4.3.1. *Dehesa* Farm

In the privately-owned *dehesa*, the contribution of the *ecosystem service* (ES) to the final products consumed, from highest to lowest in absolute values per unit area, are: private amenity, stored natural water, carbon, landscape conservation, cork, hunting, recreation, mushrooms, grazing, threatened wild biodiversity preservation and firewood (Tables 5 and S6).

In the private *dehesa* the cultural ecosystem services are notably greater than the supply and regulation-maintenance services. The cultural services (hunting, private amenity and open-access recreation) account for 40.9% of the total ecosystem services of the private *dehesa*, the provisioning services (cork, firewood, grazing, mushrooms and water) make up 32.8% and the regulation-maintenance services (landscape, biodiversity and carbon) account for 26.3% (Tables 5 and S6). The contributions of the ecosystem services corresponding to the farmer and the government are similar. Cultural services predominate in the farmer activities and regulation-maintenance services in the case of the government.

**Table 5.** Ecosystem services of the case-study large *dehesa* and forest farms in Andalusia under the AAS (2010: €/ha).

| Class | Privately-Owned *dehesa* Farm | | | Publicly-Owned *dehesa* Farm | | | Privately-Owned Forest Farm | | | Publicly-Owned Forest Farm | | |
|---|---|---|---|---|---|---|---|---|---|---|---|---|
| | WPeu | NOMeo | ES$_{ep}$ | WPeu | NOMeo | ES$_{ep}$ | WPeu | NOMeo | ES$_{ep}$ | WPeu | NOMeo | ES$_{ep}$ |
| 1. Farmer | 53.9 | 148.6 | 202.5 | 25.7 | 9.6 | 35.3 | 0.8 | 247.4 | 248.2 | 3.4 | 6.0 | 9.4 |
| 1.1 Wood forest products | 43.1 | | 43.1 | 21.0 | | 21.0 | 0.4 | | 0.4 | 1.9 | | 1.9 |
| Timber | | | | 4.5 | | 4.5 | 0.4 | | 0.4 | 1.9 | | 1.9 |
| Cork | 41.9 | | 41.9 | 16.5 | | 16.5 | | | | | | |
| Firewood | 1.3 | | 1.3 | 0.0 | | 0.0 | 0.0 | | 0.0 | | | |
| 1.2 Non-wood forest products | 10.8 | 148.6 | 159.4 | 4.7 | 9.6 | 14.4 | 0.4 | 247.4 | 247.8 | 1.5 | 6.0 | 7.5 |
| Nuts | | | | | 0.1 | 0.1 | | 2.6 | 2.6 | | 0.2 | 0.2 |
| Grazing | | 12.3 | 12.3 | | 1.3 | 1.3 | | 12.8 | 12.8 | | 0.9 | 0.9 |
| Aromatic plants | | | | | | | | | | | 0.4 | 0.4 |
| Hunting | 10.8 | 17.2 | 28.0 | 4.7 | 8.3 | 13.0 | 0.4 | 1.3 | 1.7 | 1.5 | 4.5 | 5.9 |
| Agricultural crops | | | | | | | | | | | | |
| Amenity services | | 119.1 | 119.1 | | | | | 230.6 | 230.6 | | | |
| 2. Government non-wood forest products | | 205.4 | 205.4 | | 385.1 | 385.1 | | 87.7 | 87.7 | | 249.3 | 249.3 |
| Recreation services | | 19.5 | 19.5 | | 84.9 | 84.9 | | 4.9 | 4.9 | | 67.0 | 67.0 |
| Mushrooms | | 12.7 | 12.7 | | 14.8 | 14.8 | | 12.7 | 12.7 | | 5.0 | 5.0 |
| Carbon | | 52.3 | 52.3 | | 85.3 | 85.3 | | 56.9 | 56.9 | | 57.5 | 57.5 |
| Landscape | | 44.2 | 44.2 | | 72.2 | 72.2 | | | | | 55.0 | 55.0 |
| Biodiversity | | 10.8 | 10.8 | | 18.0 | 18.0 | | 0.5 | 0.5 | | 15.8 | 15.8 |
| Water supply runoff | | 65.9 | 65.9 | | 109.9 | 109.9 | | 12.8 | 12.8 | | 48.9 | 48.9 |
| Farm (1 + 2) | 53.9 | 354.0 | 407.9 | 25.7 | 394.7 | 420.4 | 0.8 | 335.1 | 335.9 | 3.4 | 255.3 | 258.7 |

Abbreviations: WPeu is environmental work in progress used; NOMeo is ordinary environmental net operating margin; ES$_{ep}$ is ecosystem services at environmental prices.

In the case-study private *dehesa*, grazing consumption by game species exceeds that of the livestock species and 81.5% of the forage units consumed by game species have been valued at a price of zero (Table S2). Attributing the origin of the game species ecosystem service to the grazing is justified by its "free" consumption being a necessary condition of "wild" rearing. Thus, the controlled animals (game and livestock species) are the receivers of the ecosystem services of grazing embedded in the environmental lease prices estimated from the leasehold livestock grazing market and the game captures by recreational hunting operators. By considering both the lease prices and then

subtracting the manufactured costs the resource rents (ecosystem services) are obtained which account for 19.9% of the ecosystem services of the farmer activities.

The public *dehesa* regulation-maintenance services make up 41.7% of the ecosystem services, while the provisioning and cultural services only account for 35.0% and 23.3%, respectively (Tables S7). The contributions of the farmer and government to the ecosystem services of the farm are 8.4% and 91.6%, respectively. (Tables 5 and S7).

#### 4.3.2. Forest Farm

In the public forest farm the contribution of the ecosystem services (ES) to the final products consumed, in order of magnitude, are: recreation service, carbon service, landscape conservation service, stored natural water, threatened wild biodiversity preservation service, hunting service, mushroom picking, timber, grazing, collection of aromatic plants and industrial edible fruit (pine nuts) (Tables 5 and S8).

In the public forest the regulation-maintenance services make up 49.6% of the ecosystem services of the forest, while the cultural and provisioning services account for considerably less at 22.2% and 28.2%, respectively (Tables 5 and S8). The ecosystem services corresponding to the farmer account for 3.6% and those of the government 96.4% of the forest ES. Provisioning services make up the main part of the farmer ES, whereas regulation-maintenance services predominate among the government ES.

In the public forest the consumption of grazing by livestock is slightly higher than that of game species. 98.9% of the forage units consumed by the game species has been valued at a price of zero (Table S5). Thus, the ecosystem services of livestock grazing and game species captures, according to their opening inventory value, make up 73.1% of the farmer ecosystem services attributable to livestock grazing observed in the markets and of livestock species, assuming a substitutive value for the ES of the game captures (Tables 5 and S8).

In the privately-owned forest farm, the cultural ecosystem services make up 70.6% of the ecosystem services while the provisioning and the regulation-maintenance services account for 12.3% and 17.1%, respectively (Tables 5 and S9). The contribution of the farmer to these ecosystem services of the privately-owned forest is 73.9% and that of the government is 26.1%. Cultural ecosystem services are the main farmer services and regulation-maintenance services are the main ones corresponding to the government (Table 5 and S9).

#### 4.4. Contribution of Environmental Income to the Total Income of Society

The total sustainable income of society from the case-study *dehesa* and forest farms is the most important synthetic variable of all the economic indicators estimated. The total sustainable income incorporates the net value added and the gain/loss of capital (GC) at the closing of the period. Furthermore, the GC corrects the limitations of the net value added caused by the double counting of natural growth and carbon fixation. Therefore, the total income includes the income from capital, which represents the upper bound sustainable consumption of products obtained by the *dehesa* and forest farms from the total immobilized capital investment measured under the rSNA and AAS, respectively (presented in detail in Tables S10–S13 and S14–S17). The environmental asset gain (EAg) is the concept which reflects the profit from nature additional to the environmental net operating margin (NOMe) obtained by the owners due to the revaluations associated with the reduction by one period in the number of discount periods for the inventoried woody and game species environmental assets, unexpected changes of environmental prices at opening period, extraordinary destructions and instrumental adjustments.

No extraordinary destructions of capital have taken place with the exception of livestock head natural mortality, and the greater effect on the capital gain is due to the negative change of the land prices, which results in a negative value for the capital gain (loss of capital). This is not of particular importance as the investment rationale of the

owners of large silvopastoral *dehesa* and forest farms tends towards bequeathing the farms to their descendants in the case of the private owners and to society as a whole in the case of public owners. Over periods of several years the land prices of the case-study farms present moderately positive trends in real terms [16].

The capital balance account (presented in detail in Tables S18–S21) is where the instrumental adjustments are registered which avoid double counting due to depreciations of fixed capital (fixed capital consumption) considered in the estimates of net operating surpluses by rSNA and margins by AAS as well as implicitly in the closing fixed capital of the period. Additionally, the adjustments of the opening environmental values are registered for natural growth of woody products and carbon final product consumed (net fixation of destruction due to catastrophic fires anticipated in accordance with their historic rates).

4.4.1. *Dehesa* Farm

The environmental income (EI) of the private *dehesa* makes up 60.6% of the total income of 426.8 €/ha (Tables 6, 7 and S6). Cork and hunting account for 90.8% of the environmental income of 196.1€/ha from the commercial products of the private *dehesa* and exceed the negative environmental income of the farmer private amenity of −90.0 €/ha (Tables 6 and S6). In the case of the government activities the negative value for the environmental asset of carbon reduces the environmental income to a value close to zero (Tables 7 and S6). Water and landscape make up 72.2% of the government environmental income (Tables 6 and S6). The environmental incomes of the farmer activities are 30.4% lower than those of the government activities in the private *dehesa*. If we omit the effect of the variation in land prices, the contributions of the farmer activities would be greater than that of the government. The decrease in the price of land of −209.1€/ha is the main reason for the negative value of −182.0 €/ha estimated for the capital gain (Tables 7 and S6). The result at the end of the period is that the negative capital gain reduces the total income to 70.1% of the net value added at social prices in the private *dehesa* (Tables 7 and S6).

**Table 6.** Contribution of environmental income to total income of the case-study large *dehesa* and forest farms in Andalusia under the AAS (2010: €/ha).

| Class | Privately-Owned *dehesa* Farm | | | | Publicly-Owned *dehesa* Farm | | | | Privately-Owned Forest Farm | | | | Publicly-Owned Forest Farm | | | |
|---|---|---|---|---|---|---|---|---|---|---|---|---|---|---|---|---|
| | $EI_{ep}$ | $CIm_{sp}$ | LC | $TI_{sp}$ | $EI_{ep}$ | $CIm_{sp}$ | LC | $TI_{sp}$ | $EI_{ep}$ | $CIm_{sp}$ | LC | $TI_{sp}$ | $EI_{ep}$ | $CIm_{sp}$ | LC | $TI_{sp}$ |
| 1. Farmer | 106.1 | 18.7 | 106.5 | 231.3 | −102.1 | 4.2 | 30.7 | −67.3 | 91.1 | −30.9 | 48.6 | 108.9 | −131.1 | 2.9 | 21.8 | −106.4 |
| 1.1 Wood forest products | 157.6 | 5.8 | 18.6 | 182.0 | 122.8 | 2.4 | 10.2 | 135.4 | 4.1 | 0.1 | 4.9 | 9.0 | 39.8 | 0.3 | 4.2 | 44.2 |
| Timber | 1.0 | 0.4 | 0.4 | 1.8 | 12.9 | 3.2 | 6.0 | 22.1 | 4.0 | 0.1 | 4.6 | 8.7 | 38.0 | 0.3 | 4.2 | 42.4 |
| Cork | 150.7 | 2.7 | 16.1 | 169.5 | 107.1 | −0.9 | 3.9 | 110.1 | | | | | | | | |
| Firewood | 5.9 | 2.7 | 2.1 | 10.7 | 2.8 | 0.0 | 0.3 | 3.1 | 0.0 | 0.0 | 0.2 | 0.2 | 1.8 | | 0.0 | 1.8 |
| 1.2 Non-wood forest products | −51.5 | 12.8 | 88.0 | 49.3 | −225.0 | 1.8 | 20.4 | −202.7 | 87.1 | −30.9 | 43.8 | 99.9 | −170.9 | 2.7 | 17.6 | −150.6 |
| Nuts | 0.0 | | | 0.0 | 0.2 | 0.1 | 3.4 | 3.6 | 2.9 | −4.3 | 3.8 | 2.4 | 0.3 | 0.0 | 0.4 | 0.8 |
| Grazing | 11.1 | 37.1 | 14.1 | 62.3 | 1.1 | −0.1 | 0.0 | 1.1 | 12.8 | −0.1 | 1.2 | 13.9 | 1.0 | 0.2 | | 1.1 |
| Conservation forestry | | 0.1 | 0.7 | 0.8 | | 0.2 | 5.1 | 5.3 | | −0.1 | | −0.1 | | 0.9 | 11.7 | 12.6 |
| Aromatic plants | | | | | | | | | | | | | 0.4 | 0.0 | 1.5 | 1.9 |
| Hunting | 27.5 | −4.5 | 14.5 | 37.5 | 12.9 | 0.0 | 10.4 | 23.3 | 1.7 | −0.4 | 11.9 | 13.3 | 6.2 | 0.5 | 3.3 | 9.9 |
| Commercial recreation | | −0.6 | 0.0 | −0.6 | | 0.0 | 0.0 | 0.0 | | | | | | | | |
| Residential | | −25.5 | 6.9 | −18.7 | | 0.4 | | 0.4 | | −26.3 | | −26.3 | | 0.3 | | 0.3 |
| Livestock | | 0.8 | 45.2 | 46.1 | | 1.2 | 1.6 | 2.8 | | −2.8 | 21.8 | 18.9 | | 0.7 | 0.7 | 1.4 |
| Agricultural crops | | 5.4 | 6.4 | 11.8 | | 0.0 | 0.0 | 0.0 | | 3.0 | 5.1 | 8.1 | | 0.1 | 0.0 | 0.1 |
| Amenity services | −90.0 | | | −90.0 | −239.2 | | | −239.2 | 69.6 | | | 69.6 | −178.8 | | | −178.8 |
| 2. Government non-wood forest products | 152.4 | 3.0 | 40.1 | 195.5 | 316.4 | 8.6 | 54.5 | 379.5 | 25.9 | 5.8 | 28.7 | 60.4 | 215.0 | 0.6 | 49.2 | 264.8 |
| Fire services | | −2.3 | 25.1 | 22.8 | | −3.6 | 28.6 | 25.0 | | 0.2 | 19.2 | 19.4 | | −2.3 | 25.7 | 23.3 |
| Recreatrion services | 19.5 | 0.5 | 4.2 | 24.2 | 84.9 | 7.9 | 7.1 | 99.9 | 4.9 | 1.4 | 2.0 | 8.3 | 67.0 | −1.9 | 8.9 | 74.0 |

| | EIep | CImsp | LC | TIsp | EIep | CImsp | LC | TIsp | EIep | CImsp | LC | TIsp | EIep | CImsp | LC | TIsp |
|---|---|---|---|---|---|---|---|---|---|---|---|---|---|---|---|---|
| Mushrooms | 12.7 | 1.0 | 0.1 | 13.8 | 14.8 | 1.0 | 0.2 | 16.0 | 12.7 | 0.9 | 0.2 | 13.7 | 5.0 | 1.0 | 0.1 | 6.1 |
| Carbon | −0.6 | | | −0.6 | 16.6 | | | 16.6 | −4.9 | | | −4.9 | 23.2 | | | 23.2 |
| Landscape | 44.2 | 0.9 | 6.7 | 51.8 | 72.2 | 0.5 | 13.4 | 86.1 | | 0.9 | 4.0 | 5.0 | 55.0 | 0.6 | 10.2 | 65.8 |
| Biodiversity | 10.8 | 2.9 | 3.9 | 17.5 | 18.0 | 2.8 | 5.2 | 26.0 | 0.5 | 2.3 | 3.3 | 6.1 | 15.8 | 3.3 | 4.3 | 23.4 |
| Water supply runoff | 65.9 | | | 65.9 | 109.9 | | | 109.9 | 12.8 | | | 12.8 | 48.9 | | | 48.9 |
| Farm (1 + 2) | 258.5 | 21.7 | 146.6 | 426.8 | 214.3 | 12.7 | 85.2 | 312.2 | 117.0 | −25.1 | 77.3 | 169.2 | 83.9 | 3.6 | 71.0 | 158.4 |

Abbreviations: EIep is environmental income at environmental prices; CImsp is manufactured capital income at social prices; LC is labour compensation; TIsp is total income at social prices.

**Table 7.** Contribution of capital gain to total income of the case-study large *dehesa* and forest farms in Andalusia under the AAS (2010: €/ha).

| Class | Privately-Owned *dehesa* Farm | | | Publicly-Owned *dehesa* Farm | | | Privately-Owned Forest Farm | | | Publicly-Owned Forest Farm | | |
|---|---|---|---|---|---|---|---|---|---|---|---|---|
| | NVAsp | CG | TIsp | NVAsp | CG | TIsp | NVAsp | CG | TIsp | NVAsp | CG | TIsp |
| 1. Farmer | 377.9 | −146.6 | 231.3 | 76.6 | −143.9 | −67.3 | 323.8 | −214.9 | 108.9 | 41.7 | −148.2 | −106.4 |
| 1.1 Wood forest products | 52.1 | 129.9 | 182.0 | 35.1 | 100.3 | 135.4 | 5.2 | 3.8 | 9.0 | 10.6 | 33.6 | 44.2 |
| Timber | 1.1 | 0.7 | 1.8 | 10.3 | 11.8 | 22.1 | 5.0 | 3.8 | 8.7 | 10.5 | 31.9 | 42.4 |
| Cork | 45.8 | 123.7 | 169.5 | 24.4 | 85.8 | 110.1 | | | | | | |
| Firewood | 5.2 | 5.5 | 10.7 | 0.4 | 2.7 | 3.1 | 0.2 | 0.0 | 0.2 | 0.0 | 1.7 | 1.8 |
| 1.2 Non-wood forests products | 325.8 | −276.5 | 49.3 | 41.5 | −244.2 | −202.7 | 318.6 | −218.7 | 99.9 | 31.2 | −181.8 | −150.6 |
| Nuts | | 0.0 | 0.0 | 3.5 | 0.1 | 3.6 | 7.7 | −5.3 | 2.4 | 0.6 | 0.2 | 0.8 |
| Grazing | 66.2 | −4.0 | 62.3 | 1.6 | −0.5 | 1.1 | 14.6 | −0.7 | 13.9 | 1.2 | −0.1 | 1.1 |
| Conservation forestry | 0.9 | −0.1 | 0.8 | 5.4 | −0.1 | 5.3 | 0.7 | −0.7 | −0.1 | 12.9 | −0.2 | 12.6 |
| Aromatic plants | | | | | | | | | | 2.0 | 0.0 | 1.9 |
| Hunting | 46.5 | −9.0 | 37.5 | 25.7 | −2.3 | 23.3 | 23.0 | −9.7 | 13.3 | 10.9 | −1.0 | 9.9 |
| Commercial recreation | −0.6 | 0.0 | −0.6 | 0.0 | 0.0 | 0.0 | | | | | | |
| Residential | 3.4 | −22.1 | −18.7 | 0.4 | | 0.4 | 1.5 | −27.8 | −26.3 | 0.3 | 0.0 | 0.3 |
| Livestock | 75.6 | −29.5 | 46.1 | 4.9 | −2.1 | 2.8 | 31.3 | −12.3 | 18.9 | 3.2 | −1.8 | 1.4 |
| Agricultural crops | 14.6 | −2.7 | 11.8 | 0.0 | 0.0 | 0.0 | 9.1 | −1.0 | 8.1 | 0.1 | 0.0 | 0.1 |
| Amenity services | 119.1 | −209.1 | −90.0 | | −239.2 | −239.2 | 230.6 | −161.0 | 69.6 | | −178.8 | −178.8 |
| 2. Government non-wood forests products | 230.9 | −35.4 | 195.5 | 414.2 | −34.7 | 379.5 | 103.8 | −43.4 | 60.4 | 275.8 | −10.9 | 264.8 |
| Fire services | 25.1 | −2.3 | 22.8 | 28.6 | −3.6 | 25.0 | 19.2 | 0.2 | 19.4 | 25.7 | −2.3 | 23.3 |
| Recreation services | 22.6 | 1.6 | 24.2 | 95.0 | 4.9 | 99.9 | 7.2 | 1.1 | 8.3 | 77.0 | −3.0 | 74.0 |
| Mushrooms | 13.4 | 0.5 | 13.8 | 15.5 | 0.5 | 16.0 | 13.4 | 0.3 | 13.7 | 5.6 | 0.5 | 6.1 |
| Carbon | 37.5 | −38.1 | −0.6 | 55.1 | −38.5 | 16.6 | 42.7 | −47.6 | −4.9 | 32.1 | −8.9 | 23.2 |
| Landscape | 51.0 | 0.8 | 51.8 | 85.9 | 0.2 | 86.1 | 4.0 | 0.9 | 5.0 | 65.3 | 0.5 | 65.8 |
| Biodiversity | 15.5 | 2.1 | 17.5 | 24.2 | 1.8 | 26.0 | 4.5 | 1.6 | 6.1 | 21.1 | 2.3 | 23.4 |
| Water supply runoff | 65.9 | | 65.9 | 109.9 | | 109.9 | 12.8 | | 12.8 | 48.9 | | 48.9 |
| Farm (1 + 2) | 608.8 | −182.0 | 426.8 | 490.8 | −178.6 | 312.2 | 427.6 | −258.3 | 169.2 | 317.5 | −159.1 | 158.4 |

Abbreviations: NVAsp is net value added at social prices; CG is capital gain; TIsp is total income at social prices.

The environmental income of the public *dehesa* makes up 68.6% of the total income of 312.2 €/ha (Tables 6 and S7). In the publicly-owned *dehesa* the absence of final product consumed of landowner private amenities does not prevent the existence of the environmental asset gain since the hypothesis has been assumed that the public owner has the option to sell the farm under competitive market conditions. The cork and hunting activities account for 87.6% of the environmental incomes of 137.1€/ha, from the commercial products of the public *dehesa*. The environmental income from the commercial products is less than the negative environmental income from the publicly-owned *dehesa* private amenity, which is −239.2 €/ha (Tables 6 and S7). The negative value of for the environmental asset gain of carbon reduces the environmental income of this activity to a slightly positive value of 16.6 €/ha (Tables 7 and S7). Water, recreation service and landscape make up 84.4% of the environmental income of 316.4 €/ha for the government activities (Tables 6 and S7).

In the public *dehesa*, the negative environmental income for the aggregate individual activities of the farmer and the positive values for the government activities, explain the fact that the environmental income of 214.3 €/ha is 54.9% of the environmental net operating margin of 390.6 €/ha (Tables 6 and S7).

If we omit the effect of the variation in land prices, the aggregate environmental income for the public *dehesa* reaches an absolute value of 453.5 €/ha, which is more than double the estimated value assuming that the public *dehesa* can be sold on the competitive land market. The decrease in the price of land is the main reason for the negative value of −178.6 €/ha estimated for the capital gain (Tables 7 and S7). The result at the end of the period is that the negative capital gain reduces the total income to 63.6% of the net value added of the public *dehesa* at social prices (Table 7).

### 4.4.2. Forest Farm

As described in the case of the public *dehesa,* the absence of the private amenity final product consumed of landowner does not prevent the existence of the environmental asset gain in the public forest farm since the hypothesis has been assumed that the public owner has the option to sell the farm under competitive market conditions. The environmental income of the publicly-owned forest farm makes up 52.9% of the total income of 158.4 €/ha (Tables 6 and S8). Timber and hunting account for 92.7% of the environmental income of 47.7 €/ha for the commercial products of the public forest farm. The value of these commercial products is less than the negative environmental income of the farmer private amenities of −178.8 €/ha, which is why the environmental income of the farmer is negative, at −131.6 €/ha (Tables 6 and S8). The recreation service, landscape and water make up 79.5% of the total environmental income of the government of 215.0 €/ha (Tables 6 and S8). The negative environmental income for the aggregate individual activities of the farmer and the positive values for the government activities, explain the fact that the environmental income of 83.9 €/ha is 35.2% of the environmental net operating margin of 238.2 €/ha (Tables 6 and S8). If we omit the effect of the variation in the prices of land, the aggregate environmental income for the public forest reaches an absolute value of 262.7 €/ha, which is 3.1 times greater than the estimated value assuming that the public *dehesa* can be sold on the competitive land market. The drop in the price of land is the main reason for the negative value of −159.1 €/ha estimated for the capital gain (Tables 7 and S8). The result at the end of the period is that the negative capital gain reduces the total income to 49.9% of the net value added of the public forest farm (Tables 7 and S8).

The environmental income of the private forest accounts for 69.1% of the total income, which is 169.2 €/ha (Tables 6 and S9). Timber and grazing make up 78.2% of the environmental income of the commercial products of the private forest, with a value of 21.5 €/ha. This environmental income is slightly lower than the environmental income of the farmer private amenity of 69.6 €/ha (Tables 6 and S9). Water and mushrooms make up 98.2% of the environmental income of the government (Tables 6 and S9). The value of the environmental income of landscape is zero and that of carbon is moderately negative (Tables 6 and S9). The environmental income of the activities of the farmer is 3.5 times greater than that of the government. The fall in the price of land of −161.0 €/ha underlies the negative value of −258.3€/ha estimated for the capital gain (Table 7 and S9). The result at the end of the period is that the negative capital gain reduces the total income to 39.6% of the net value added of the privately-owned forest farm.

### 4.5. Profitability Rates under AAS and rSNA

#### 4.5.1. Profitability Rates under AAS

The operating profitability rates (Po) of the wood products measured by the AAS vary between the lower bound of 0.9% and the upper bound of 1.8% in the public forest farm and the private *dehesa*, respectively (Table 8). The total profitability rates (P) of the wood forest products are significantly higher than those of the Po, with the lower bound of 4.4% and the upper bound of 8.7% in the private forest and private *dehesa*, respectively (Table 8).

As for the economic activities of the farms as a whole, the AAS measures operating profitability rates of 2% at the lower bound and 4.9% at the upper bound for the public and private forest farms, respectively (Table 8). Total farm profitability rates vary between a lower bound of 0.7% and an upper bound of 2.2% for public forest and private *dehesa* farms (Table 8).

*Dehesa* Farm

In the private *dehesa* the operating profitability rate (Po) of the farmer may approximately reflect a minimum of the long-term real total profitability rate (Table 8). In 2010, the negative variations in the market prices of the construction, equipment and land led to negative capital gain profitability rates (Pg) for the economic activities, with the exception of activities with work in progress due to the effect of the revaluations and the limited employment of inanimate manufactured fixed capital (Table 8). The profitability of the private *dehesa* mainly comes from the private amenity and landscape activities. The private amenity activity demands as inputs, ordinary own non-commercial intermediate consumption of private amenity services (SSncooa) to a value of 178.9 €/ha (Table S10). The landscape activity demands non-commercial intermediate products of compensation services (ISSncc) to a value of 37.9 €/ha (Table S10).

The total profitability rate of the farmer is notably lower than the total for the private *dehesa* as a whole. The total profitability rate for the government activities is greater than the total profitability rate for the private *dehesa*. These results are mainly due to the lower influence of the estimated capital gains (Table 8). The results for the farmer and government activities as a whole reveal that the observed prices in the market and simulated transactions give competitive operating profitability rates for the farmer and government. For all the activities as a whole, the aggregate operating and total (P) profitability rates are 3.7% and 2.2%, respectively (Table 8).

The operating (Po) and total (P) profitability rates of the public *dehesa* are lower than those of the private *dehesa*. In this case it is due to the absence of the private amenity service final product consumed, and to the effect of lower profitability rates of negative capital gain (Pg) (Table 8). Details for immobilized capital by activity on *dehesas* with livestock presence by activity in Andalusia can be found in Tables S22, S23 and S26.

**Table 8.** Profitability rates of the case-study large *dehesa* and forest farms in Andalusia under the AAS and rSNA (2010: %).

| Class | Privately-Owned *dehesa* Farm | | | Publicly-Owned *dehesa* Farm | | | Privately-Owned Forest Farm | | | Publicly-Owned Forest Farm | | |
|---|---|---|---|---|---|---|---|---|---|---|---|---|
| | Po | Pg | P | Po | Pg | P | Po | Pg | P | Po | Pg | P |
| Agroforestry Accounting System (AAS) | | | | | | | | | | | | |
| 1. Farmer | 3.5 | −1.9 | 1.6 | 0.6 | −2.0 | −1.3 | 4.8 | −3.7 | 1.0 | 0.4 | −2.7 | −2.3 |
| 1.1 Wood products | 1.8 | 6.9 | 8.7 | 1.4 | 5.8 | 7.2 | 0.3 | 4.1 | 4.4 | 0.9 | 5.0 | 5.9 |
| Timber | 2.9 | 2.9 | 5.7 | 3.2 | 8.8 | 12.0 | 0.3 | 4.1 | 4.5 | 1.0 | 5.0 | 6.0 |
| Cork | 1.7 | 7.1 | 8.8 | 1.3 | 5.5 | 6.8 | | | | | | |
| Firewood | 2.8 | 5.0 | 7.9 | 0.3 | 5.7 | 6.0 | −2.2 | 4.0 | 1.8 | 0.1 | 5.1 | 5.2 |
| 1.2 Non-wood forest products | 4.1 | −4.8 | −0.7 | 0.4 | −4.4 | −4.0 | 4.8 | −3.9 | 1.0 | 0.3 | −3.8 | −3.5 |
| Nuts | 0.0 | 2.8 | 2.8 | 2.8 | 1.7 | 4.6 | 6.9 | −9.2 | −2.3 | 3.0 | 3.0 | 6.0 |
| Grazing | 5.3 | −0.4 | 4.9 | 0.2 | −0.1 | 0.1 | 1.4 | −0.1 | 1.3 | 0.2 | 0.0 | 0.2 |
| Conservation forestry | 3.5 | −1.5 | 2.0 | 3.0 | −0.6 | 2.4 | 3.0 | −3.2 | −0.2 | 3.0 | −0.6 | 2.4 |
| Aromatic plants | | | | | | | | | | 3.1 | −0.3 | 2.8 |
| Hunting | 5.8 | −1.6 | 4.1 | 5.5 | −0.8 | 4.7 | 3.2 | −2.8 | 0.4 | 6.2 | −0.8 | 5.4 |
| Commercial recreation | −3.4 | 0.2 | −3.2 | −10.9 | −2.8 | −13.8 | | | | | | |
| Residential | −0.6 | −4.0 | −4.6 | 1.4 | 0.0 | 1.4 | 0.3 | −5.2 | −4.9 | 3.9 | −0.1 | 3.8 |
| Livestock | 3.9 | −3.8 | 0.1 | 5.9 | −3.8 | 2.1 | 3.0 | −3.9 | −0.9 | 5.2 | −3.7 | 1.5 |
| Agricultural crops | 7.7 | −2.6 | 5.1 | −0.7 | −0.1 | −0.8 | 5.0 | −1.2 | 3.7 | 0.3 | 0.0 | 0.3 |
| Amenity services | 4.2 | −7.4 | −3.2 | 0.0 | −5.7 | −5.7 | 6.9 | −4.8 | 2.1 | 0.0 | −4.6 | −4.6 |
| 2. Government non-wood forest products | 4.0 | −0.7 | 3.3 | 3.8 | −0.4 | 3.4 | 5.5 | −3.2 | 2.3 | 3.3 | −0.2 | 3.1 |
| Fire services | 0.0 | −3.2 | −3.2 | 0.0 | −4.4 | −4.4 | 0.0 | 0.3 | 0.3 | 0.0 | −3.2 | −3.2 |
| Recreation services | 2.8 | 0.1 | 2.9 | 2.9 | 0.1 | 3.0 | 2.8 | 0.1 | 2.9 | 2.7 | 0.2 | 2.9 |
| Mushrooms | 2.8 | 0.1 | 2.9 | 2.9 | 0.1 | 3.0 | 2.8 | 0.1 | 2.9 | 2.7 | 0.2 | 2.9 |

| | Po | Pg | P | Po | Pg | P | Po | Pg | P | Po | Pg | P |
|---|---|---|---|---|---|---|---|---|---|---|---|---|
| Carbon | 7.5 | −7.7 | −0.1 | 6.3 | −4.4 | 1.9 | 12.1 | −13.4 | −1.4 | 4.3 | −1.2 | 3.1 |
| Landscape | 3.0 | 0.1 | 3.0 | 3.0 | 0.0 | 3.0 | 0.0 | 18.7 | 18.7 | 3.0 | 0.0 | 3.0 |
| Biodiversity | 2.8 | 0.5 | 3.3 | 2.9 | 0.3 | 3.1 | 1.9 | 2.4 | 4.3 | 2.8 | 0.4 | 3.2 |
| Water supply runoff | 5.8 | 0.0 | 5.8 | 6.4 | 0.0 | 6.4 | 6.0 | 0.0 | 6.0 | 4.2 | 0.0 | 4.2 |
| AAS farm (1 + 2) | 3.7 | −1.5 | 2.2 | 2.4 | −1.1 | 1.4 | 4.9 | −3.6 | 1.3 | 2.0 | −1.3 | 0.7 |
| Refined System of National Accounts (rSNA) | | | | | | | | | | | | |
| 1. Farmer | −1.1 | 1.3 | 0.2 | −0.5 | 3.0 | 2.5 | −3.6 | −2.2 | −5.8 | 0.1 | 2.0 | 2.1 |
| 1.1 Wood forest products | 1.8 | 6.9 | 8.7 | −0.4 | 5.8 | 5.4 | −26.2 | 4.1 | −22.1 | 0.2 | 5.0 | 5.2 |
| 1.2 Non-wood forest products | −2.9 | −2.3 | −5.2 | −0.7 | −0.4 | −1.1 | −2.7 | −2.5 | −5.1 | 0.1 | −0.3 | −0.3 |
| 2. Government non-wood forest products | 3.9 | 0.2 | 4.1 | 4.4 | 0.2 | 4.5 | 3.1 | 0.5 | 3.6 | 3.0 | −0.1 | 2.8 |
| rSNA farm (1 + 2) | 0.2 | 1.0 | 1.2 | 1.6 | 1.8 | 3.4 | −2.0 | −1.5 | −3.5 | 1.6 | 0.9 | 2.5 |

Abbreviations: Po is operating profitability rate; Pg is capital gain profitability rate; P is total profitability rate.

Forest Farm

The profitability rate of the public forest farm mainly comes from the government activities (Table 8). The aggregate profitability rate of the government activities is notably greater than that of the farmer. The result as regards the aggregate products of the government activities is a competitive operating profitability rate (Po), although the total profitability rate is not competitive (Table 8).

In the case of the private forest, the profitability rate for the government activities is slightly higher than that for the farmer activities. The operating profitability rates (Po) for both the farmer and government are higher than assumed baseline competitive operating profitability at 4.8% and 5.5%, respectively. The total profitability rates (P) are not competitive (Table 8).

Details for immobilized capital by activity on forest farms with livestock presence by activity in Andalusia can be found in Tables S24, S25 and S26.

4.5.2. Comparing Profitability Rates under the rSNA and AAS

The operating profitability rates for the wood products of the privately-owned *dehesa* farm estimated by the rSNA and AAS coincide, while in the case of the publicly-owned forest farm the AAS estimates are significantly higher (Table 8). The total profitability rates for wood products of privately-owned *dehesa* farms estimated by the rSNA and AAS coincide, while for the publicly-owned forest farm the AAS estimates are slightly higher (Table 8).

For all the activities of the case-study farms, the operating profitability rates under the AAS for the privately-owned *dehesa* significantly exceed those estimated by the rSNA (Table 8). In contrast, the total profitability rates for activities measured by the AAS are significantly lower than those estimated by the rSNA (Table 8).

The reason for the difference in the operating profitability rate measurements under the AAS and rSNA for privately-owned *dehesa* farms is due to the incorporation of the environmental margin of the private amenity in the AAS. The total profitability rate for the publicly-owned forest farm estimated under the rSNA is different from that estimated by the AAS due to the inclusion of own ordinary non-commercial intermediate consumption of services in the AAS (Table 8).

*4.6. Comparing Incomes and Ecosystem Services under the rSNA and AAS*

The gross values added (GVA) differences between the rSNA and AAS in this research are due exclusively to the estimation of the gross operating surplus (GOS) in the rSNA and the gross operating margin (GOM) in the AAS. The final products of services without market prices consumed, measured under the AAS account for 78.9% and 78.5% of the net values added of the case-study private *dehesa* and the public forest farms, respectively.

Livestock rearing in the private *dehesa* and public forest farms makes up 30.9% and 0.9% respectively, of the labour compensation for the farm activities as a whole measured by the rSNA and AAS.

The ecosystem services measured by the AAS for the private *dehesa* and public forest are more than 4.6 and 2.7 times those measured by the rSNA, respectively (Table 9).

The total incomes measured by the AAS for the private *dehesa* and public forest farms are more than 2.6 and 1.3 times the net values added estimated by the rSNA, respectively.

The environmental incomes measured by the AAS for the private *dehesa* and public forest farms make up 60.6% and 52.9% of their respective total incomes. The aggregate landscape and water activities incorporate 61.8% and 41.3% of the environmental incomes of the government in the *dehesa* and forest farms, respectively.

The rSNA and AAS results at basic and at social prices, respectively, are described below in the form of gross values added (GVA) and ecosystem services (ES). We omit the comparison of total incomes as these are not incorporated in the rSNA (Tables 9, S10–S13, S14–S17). However, it would be possible from the rSNA production and capital balance accounts to also simulate the measurement of total income of the ISIC commercial products of the farmer registered in the case-study *dehesa* and forest farms.

### 4.6.1. Gross Values Added

The AAS methodology proves the consistency of the estimation of values added at social prices for the aggregate activities of the case-study *dehesa* and forest farms. The gross values added obtained at social prices coincide with the observed and simulated market transaction values at producer and basic prices (Table 9). Under the rSNA, the producer prices and basic prices also coincide. However, the values added of the individual activities of the farmer and the government can be affected by the type of price applied.

The comparisons of the estimates of gross values added under the AAS and rSNA reveal notable differences with changes in the type of price applied to the common activities and the incorporation of the carbon activity in the AAS. The main cause of these differences is the substitution of the cost prices in the rSNA for the simulated transaction prices applied in the AAS to the final products without market prices consumed.

### *Dehesa* Farm

In the case of the private *dehesa*, the AAS estimates of gross values added (GVA) of the livestock and hunting activities reveal that the application of social prices multiplies the results obtained at basic prices by 4.2 and 10.3, respectively (Table 9). The GVA of the individual activities of the government at social and basic prices do not change (Table 9). Neither does the results for the GVA corresponding to the farmer and to the farm as a whole estimated at social or basic prices.

The comparison of the GVA for livestock and hunting activities at social prices in the AAS and at basic prices in the rSNA reveals that the AAS estimates values of 92.3 €/ha and 53.5 €/ha, which contrast with the rSNA results of −24.4 €/ha and 5.1 €/ha, respectively (Table S14). The GVA of the recreation and landscape conservation service activities estimated at the same prices by the AAS are 4.3 and 5.9 times, respectively, those estimated by the rSNA (Table 9). There are notable differences in the GVA at social prices under the AAS and at basic prices under the rSNA for the aggregate activities of the farmer, the government and the private *dehesa* farm as a whole, the AAS values being 4.3, 2.2 and 3.2 times greater, respectively (Table 9).

### Forest Farm

In the public forest, the AAS estimates of the GVA of livestock, hunting and aggregate activities of the farmer show than the results at social prices are 2.2, 2.5 and 1.4 times, respectively, the values at social prices (Table 9). The GVA of the landscape conservation service activity is slightly lower at basic prices than at social prices as the latter incorporates own ordinary non-commercial intermediate consumption of public owner

donations (SSncood) (Table 9). However, the slight change in the landscape activity is insufficient to change the results for the GVA of the government and public forest farm as a whole estimated at social or basic prices (Table 9).

The comparisons of the GVA estimates for the livestock and hunting activities at social prices in the AAS and basic prices in the rSNA show that the AAS estimates values 6.0 and 3.2 times greater, respectively, than those of the rSNA (Tables 9 and S17). The GVA estimated at the same prices for the recreation and landscape conservation activities under the AAS are 7.7 and 5.4 times greater, respectively, than those estimated by the rSNA (Table 9). There are notable differences in the GVA at social prices under the AAS and at basic prices in the rSNA for the aggregate activities of the farmer, the government and the public forest farm as a whole, the AAS values being 1.7, 3.0 and 2.7 times greater than those of the rSNA, respectively (Table 9).

### 4.6.2. Ecosystem Services

*Dehesa* Farm

The value for the ecosystem service (ES) of the private amenity activity in the private *dehesa* measured, the product consumed, at social prices under the AAS is 60% lower than its value at basic prices. The rSNA values the ES of activities without market prices at a price of zero, which is the case of the amenity, recreation, landscape and threatened biodiversity activities (Table 9).

The ecosystem service values for the aggregate activities of the farmer, the government and the private *dehesa* farm, as a whole, presents notable differences depending on whether the social prices of the AAS product consumed or the basic prices of the rSNA is applied. Under the AAS, the ES values for the farmer, the government and the private *dehesa* as a whole are 2.4, 3.0 and 2.7 times greater, respectively, than those of the rSNA (Table 9).

**Table 9.** Comparison of gross value added and ecosystem services of the case-study large *dehesa* and forest farms in Andalusia under the AAS and the rSNA (2010).

| Class | Privately-Owned *dehesa* Farm | | | | | | Publicly-Owned Forest Farm | | | | | |
|---|---|---|---|---|---|---|---|---|---|---|---|---|
| | $\frac{GVA_{sp,AAS}}{GVA_{bp,AAS}}$ | $\frac{GVA_{sp,AAS}}{GVA_{pp,AAS}}$ | $\frac{GVA_{sp,AAS}}{GVA_{bp,rSNA}}$ | $\frac{ES_{ep,AAS}}{ES_{bp,AAS}}$ | $\frac{ES_{ep,AAS}}{ES_{pp,AAS}}$ | $\frac{ES_{ep,AAS}}{ES_{bp,rSNA}}$ | $\frac{GVA_{sp,AAS}}{GVA_{bp,AAS}}$ | $\frac{GVA_{sp,AAS}}{GVA_{pp,AAS}}$ | $\frac{GVA_{sp,AAS}}{GVA_{bp,rSNA}}$ | $\frac{ES_{ep,AAS}}{ES_{bp,AAS}}$ | $\frac{ES_{ep,AAS}}{ES_{pp,AAS}}$ | $\frac{ES_{ep,AAS}}{ES_{bp,rSNA}}$ |
| 1. Farmer | 1.0 | 1.1 | 4.3 | 0.5 | 0.5 | 2.4 | 1.4 | 1.6 | 1.7 | 1.0 | 1.0 | 1.0 |
| 1.1 Wood forest products | 1.5 | 1.5 | 1.0 | 1.0 | 1.0 | 1.0 | 1.1 | 1.1 | 1.9 | 1.0 | 1.0 | 1.0 |
| Timber | 2.6 | 2.6 | 3.5 | | | | 1.1 | 1.1 | 1.9 | 1.0 | 1.0 | 1.0 |
| Cork | 1.5 | 1.5 | 1.0 | 1.0 | 1.0 | 1.0 | | | | | | |
| Firewood | 1.3 | 1.3 | 1.0 | 1.0 | 1.0 | 1.0 | 1.0 | 1.0 | | | | |
| 1.2 Non-wood forest products | 1.0 | 1.1 | 8.2 | 0.5 | 0.5 | 4.0 | 1.5 | 1.8 | 1.6 | 1.0 | 1.0 | 1.0 |
| Nuts | | | | | | | 1.0 | 1.0 | 1.0 | 1.0 | 1.0 | 1.0 |
| Grazing | 2.5 | 2.6 | 2.5 | 1.0 | 1.0 | 1.0 | 1.1 | 1.1 | 1.1 | 1.0 | 1.0 | 1.0 |
| Conservation forestry | 1.4 | 1.4 | 1.4 | | | | 1.1 | 1.1 | 1.1 | | | |
| Aromatic plants | | | | | | | 1.0 | 1.0 | 1.0 | 1.0 | 1.0 | 1.0 |
| Hunting | 10.3 | 10.3 | 10.6 | 1.0 | 1.0 | 1.0 | 2.5 | 2.5 | 3.2 | 1.0 | 1.0 | 1.0 |
| Commercial recreation | 1.0 | 1.0 | 1.0 | | | | | | | | | |
| Residential | 1.0 | 1.0 | 1.0 | | | | 1.0 | 1.0 | 1.0 | | | |
| Livestock | 4.2 | −7.1 | −3.8 | | | | 2.2 | −1.9 | 6.0 | | | |
| Agricultural crops | 1.0 | 1.1 | 0.9 | | | | 1.0 | 1.0 | 1.0 | | | |
| Amenity services | 0.4 | 0.4 | | 0.4 | 0.4 | | | | | | | |
| 2. Government non-wood forests products | 1.0 | 0.9 | 2.2 | 1.0 | 0.8 | 3.0 | 1.0 | 1.0 | 3.0 | 1.0 | 0.9 | 5.3 |
| Fire services | 1.0 | 1.0 | 1.0 | | | | 1.0 | 1.0 | 1.0 | | | |
| Recreation services | 1.0 | 1.0 | 4.3 | 1.0 | 1.0 | | 1.0 | 1.0 | 7.7 | 1.0 | 1.0 | |
| Mushrooms | 1.0 | 1.0 | 1.0 | 1.0 | 1.0 | 1.0 | 1.0 | 1.0 | 1.0 | 1.0 | 1.0 | 1.0 |
| Carbon | 1.0 | 1.0 | | 1.0 | 1.0 | | 1.0 | 1.0 | | 1.0 | 1.0 | |
| Landscape | 1.0 | 0.6 | 5.9 | 1.0 | 0.5 | | 0.9 | 0.8 | 5.4 | 0.9 | 0.8 | |
| Biodiversity | 1.0 | 1.0 | 3.7 | 1.0 | 1.0 | | 1.0 | 1.0 | 4.4 | 1.0 | 1.0 | |
| Water supply runoff | 1.0 | 1.0 | 1.2 | 1.0 | 1.0 | 1.2 | 1.0 | 1.0 | 1.2 | 1.0 | 1.0 | 1.2 |
| Farm (1 + 2) | 1.0 | 1.0 | 3.2 | 0.7 | 0.7 | 2.7 | 1.0 | 1.0 | 2.7 | 1.0 | 0.9 | 4.6 |

Abbreviations: GVA is gross value added; ES is ecosystem services; ep is environmental prices; sp is social prices; bp is basic prices; pp is producer prices; rSNA is revised System of National Accounts; AAS is Agroforestry Accounting System.

Forest Farm

In the case of the public forest, the AAS valuations at social and basic prices of products consumed, the ecosystem services valuations of the landscape conservation service and threatened wild biodiversity preservation service activities coincide (Table 9). The rSNA valuation of the mushroom activity ecosystem service coincides with that of the AAS. The ES of water with market prices is 20% higher when estimated by the AAS then by the rSNA (Table 9). The difference is due to the assumption in the rSNA that the water consumed by manufacturing industry, services and households only incorporates manufactured operating surplus.

Under the AAS at social prices valuations of products consumed, the ecosystem service values for the aggregate activities of the government and the public -forest as a whole are 5.3 and 4.6 times, respectively, those of the rSNA at basic prices valuations of products consumed, while the values for the ES of the farmer activities coincide under the two methodologies (Table 9).

## 5. Discussion

The natural biological productivity associated with natural growth and grazing fodder in the *dehesa* and forest farms is strongly conditioned by the variability in the bio-physical characteristics of the environment, which have not been taken into account in this study. The discussion in this case focuses on comparing the economic results for the case-study *dehesa* and forest farms. The economic results provide valuable information on which to base not only the management of the farms by the owners but also the design and implementation of public policies. Among these policies, those which address landscape conservation service, and the preservation service of threatened wild natural variety are of particular relevance nowadays. Particular emphasis has been focused on the discussion of physical production of grazing, values added and the ecosystem services of the economic activities of the case-study *dehesa* and forest farms.

### 5.1. Trade-off between Grazing and Natural Regeneration of Wood Forests Products

Extensive livestock grazing has scarce impact on the regeneration of trees of the *Pinus* genus, although it does affect plants of the *Quercus* genus and other herbaceous vegetation. In recent decades the impact of traditional grazing of livestock species has intensified along with the natural expansion and re-introduction of large game species in the case-study private *dehesa*, where the tree canopy covers 83% of the farm area, accentuating the lack of new recruitment from natural regeneration.

Government policies in Spain have generally been aimed at dissuading extensive livestock rearing in Mediterranean conifer forests in order to favour natural regeneration and protect young plantations. At the same time, the Spanish governments, in the past, had actively incentivized deforestation in species of the *Quercus* genus in order to "improve" natural productivity of grazing fodder in the *dehesa*. Even today, there is a continued absence of government policy to compensate concerted action to temporarily restrict grazing in the *dehesa*.

The application of agro-environmental measures in the private *dehesa* under the Common Agricultural Policy (CAP), aimed at the conservation of trees of the *Quercus* species and the mitigation of biodiversity loss is based in setting upper bound livestock stocking rate. In practice the lack of oak trees regeneration is not the livestock stocking rate intensity but rather the absence of temporary exclusion of livestock grazing in areas with programmed tree regeneration [30]. In addition to livestock grazing, grazing of large game species in the *dehesas* using wire-fenced enclosure can mean catastrophic degradation of the bushy vegetation of young trees and scrub.

### 5.2. Values Added Shortcomings Mitigation Challanges

In the description of results, the limitations of the measurements of the values added of the rSNA and AAS methodologies applied in the case study of *dehesa* and forest have been highlighted.

The limitation of the rSNA valuation of the gross operating surpluses at basic prices is due to the omission of the environmental work in progress used (WPeu) in the intermediate consumption and the natural growth (NG) at the close of the period in the gross capital formation (GCF) of wood forest products.

The AAS does not solve the problem of measuring environmental gross operating margins biases due to the incorporation of natural growth and the simulated final product consumed of carbon sequestration.

In the case-study *dehesa* and forest farms, an element that differentiates the respective values added to a large extent is the private amenity self-consumed only by the private owners. As the public owners of the *dehesa* and forest farms are legal (virtual) entities, they cannot self-consume private amenities. Another forestry conservation activity related aspect to be borne in mind is that forests re-naturalized through natural regeneration processes subsequent to the historical plantation, more recent planned public reforestations of stone pine and the mountain orography provide environmental services; these being production factors with a notable contribution to the net values added of the recreation and landscape conservation services of the forest farms. These net values added are notably higher than those of the private *dehesa*, which are located in lowland areas of rolling hills or plains.

In terms of the contribution of nature, the net value added is highly important because if we admit the technical f function in the total product equation (see Equation (1)), it shows (prior to introducing the price system in the f function) all the appropriate production factors for which a monetary numeraire equal to or greater than zero could be incorporated. If the environmental production factors of an individual total product consumed, j, have a price of zero, then the ecosystem service does not contribute to the economic value of the product. However, its environmental biological function informs us that the environmental production factors are the prior condition to the existence of the manufactured economic value of the product consumed. Thus, the labour compensation and manufactured investment cost which give rise to the economic value of the products consumed of the silvopastoral *montes* farms are fully reflected in the total product consumed and not by the economic ecosystem services. This is due to the fact that the existence of these ecosystem services, given consumer preferences, is related to the ownership institutions and other local institutional factors. A transaction value of zero rarely contributes much to the value of a product consumed, the existence of which may be due precisely to the fact that the market does not give an exchange value to environmental production factors.

### 5.3. Ecosystem Services

The ecosystem service values in the case-study *dehesa* and forest farms are registered for 14 activities. It is only in the case of agricultural crop products that the values do not reach the minimum exchange value of 0.1€/ha necessary to be registered. The residential service, recreation service, livestock, conservation forestry service and forest firefighting service activities do not register environmental production factors in this research.

The differences in the ecosystem service estimates of the AAS and rSNA are mainly due to the application of cost prices and social prices, respectively, in the estimation of final products without market prices consumed, and to the omission of the carbon activity by the rSNA.

The decisive factor underlying the difference in the absolute value of the ecosystem services per unit area of the large case-study *dehesa* and forest farms is the absence of

self-consumption of private amenity services in the publicly-owned *dehesa* and forest farms. This absence of private amenity self-consumption in the public *dehesa* is partially counterbalanced by the greater production of open-access recreation and carbon services in comparison to the private *dehesa*.

*5.4. Comparing Incomes and Ecosystem Services under the rSNA and AAS*

As reported in the results for the case-study *dehesa* and forest farms, the rSNA estimates omit most of the income of both the farmer and government if compared to the AAS estimates. The differences in the results obtained in the applications of the rSNA and AAS methodologies to the case-study *dehesa* and forest farms evidence the fact that the values of the incomes and ecosystem services are poorly reflected in the rSNA measurements. In contrast, the subjective measurements of the AAS methodology in the application to the case-study *dehesa* and forest farms have been shown to be consistent with the concept of total sustainable income of society.

*5.5. Policy Implications for Overcoming Refined Standard Economic Accounts of Society Shortcoming Applied to dehesa and Forest Farms*

In the *dehesa* and forest farms there is substantial demand for self-consumption of private amenities, which is only taken advantage of in the privately-owned farm. Since the private amenity service is not applicable in the public *dehesa* and forest farms, the larger production of open-access recreation and carbon services may potentially compensate for the absence of private amenity production. Thus, from the perspective of cultural ecosystem service production, the possible change in the type of ownership of a farm could be positive or negative where, in practice, there is an exchange between private amenities and open-access recreation services.

The refined standard economic accounts of society (rSNA) presents the measurement of the value added of the International Standard Industries Classification (ISIC) economic activities at national level as equivalent to the total income. This is inconsistent with the theoretical concept of national income acknowledged in the official methodological guidelines for the economic accounts of agriculture and forestry, which estimate the income from products in the ISIC list (codes 01 and 02) of the European Union [3]. The inconsistencies of the rSNA are due in part to the fact that it does not recognize the environmental assets as production factors when they lack manufactured production factors, as well as to the arbitrary assumption that the final products without market products consumed generate net operating surpluses with a value of zero. These politically based accounting criteria are usually justified by the uncertainty of simulating market transactions for products which, since their real transactions are not observable, must be obtained through procedures of stated marginal willingness to pay by real consumers or revealed behaviours in the consumption of commercial products, from transactions of flows or of assets [31,32].

However, the criteria in the rSNA methodology for excluding simulated consumer preference prices from the transaction prices system are not fulfilled in the rSNA. This is the case for the estimation of the values added of the farmer and government institutional sectors. In practice, the rSNA simulates that people value the products without market prices consumed according to their cost prices, and therefore generate labour compensation as the only component of the net value added. The inconsistency here of the rSNA is that it accepts the simulation of imputing a value of zero for the net operating surplus of consumption of products without market prices. While the rSNA rejects the valuation of these products consumed (the real value of which may be positive or negative) derived from peoples' marginal willingness to pay, this valuation is applied in the AAS. Summing up, the application of cost prices in the rSNA in place of the market prices of products is both questionable and inconsistent with consumer exchange value theory. Although the AAS application of the transaction value for the consumption of products without market prices derived from the stated or simulated prices according to

peoples' stated marginal willingness to pay may also be questionable, it is not inconsistent with income theory.

The rSNA and AAS applications to the case-study *dehesas* and forest farms reveal the modest economic numbers of nature according to the rSNA results in comparison to the AAS measurements. The comparisons of the gross values added and ecosystem services under the rSNA and AAS methodologies evidence the disadvantages of applying the cost prices in the valuations of products without market prices in the *dehesa* and forest farms.

The lesson learned from the results of this case study is that the hidden numbers of nature in the standard System of National Accounts for environmental incomes and environmental assets implies an important lack of information for the design and implementation of public policies on woodland and forest landscapes.

The incorporation of the environmental asset of private amenity is recognised in Spain under land law, which in the case of government purchase of a *dehesa* or forest farm allows the payment of a maximum value of twice the present discounted value (NPV) of the resource rents (economic rents) of the market products belonging to the landowner [33,34]. The problem with estimating a fair value for government purchase of rural land is that the government's subjective choice of the coefficient that multiplies the NPV of commercial products resource rent is a source of potential bias in the offer price determined by the government. The government should make use of algorithms generated by economic science to determine through hedonic and/or landowner-stated methods the fair value of the private amenity environmental asset of the farms. In this research this bias has been overcome by estimating the environmental asset of private amenity through the contingent valuation method applied to private non-industrial land owners of *dehesa* and forest farms in Andalusia [35].

## 6. Conclusions

This research demonstrates the viability of measuring the many individual and aggregate incomes and ecosystem services of the case-study Mediterranean *monte* farms under the Agroforestry Accounting System (AAS), although these measurements are subject to a greater number of subjective criteria than those incurred under the slightly refined standard System of National Accounts (rSNA). The extensions of the AAS simulated social prices measurements show that the subjective criteria in the rSNA of simulating zero values for the operating surpluses of the final products without market prices consumed and the omission of activities which do not incur manufactured costs lead to inconsistent valuations of the income and ecosystem services of the case-study Mediterranean *dehesa* and forest farms.

The discrepancies between the AAS and rSNA methodologies applications are not due to differences in the concepts of income and capital (except in the case of products without manufactured costs) but rather to government statistical conventions for national/regional and farm scale income statistics. The AAS results show that it is possible to extend the criterion of simulated market transaction prices for intermediate and final products without market prices consumed. This extension is consistent with the same criterion of exchange value for commercial products registered by the rSNA. Hence, it is the circumstances of time, place and demand of the private owners and consumers of public products without market prices, which will explain the negative or positive variations in the cultural ecosystem services which can occur through changes in the type of ownership and price applied to the products. Although it should be considered with caution given the lack of statistical consistency among the private and public farms, the corollary of this conclusion from the case study is that the advantage of public ownership of the farms may be linked to the preservation of unique biological variety in danger of extinction, which is reasoned beyond the price system. The advantages of private ownership, however, may be reflected by greater intensity of manufactured investment, labour compensations and the offer of private amenity environmental margins.

The United Nations Statistics Division (UNSD), in its final draft version ([1]: https://seea.un.org/ecosystem-accounting (accessed on 11 March 2021)) of the System of Environmental-Economic Accounting—Ecosystem Accounting (SEEA EA), has adopted chapters 1–7 as an international statistical standard. In the same document, chapters 8–11 present internationally recognized statistical principles and recommendations for observed and simulated market valuation of ecosystem services and environmental assets. This research has demonstrated that the available scientific knowledge allows for consistent measurements of ecosystem services and changes in environmental assets. The SEEA EA methodology agrees with and recommends the principles of valuations based on observed and simulated exchange values of final products consumed with and without market prices. This research values, beyond the SEEA EA recommendations, environmental incomes which integrate ecosystem services and changes in environmental assets in a single variable in a way that is consistent with SEEA EA valuations. Future agreements on the standard monetary SEEA EA must overcome the challenge of incorporating an indicator of environmental income.

*Dehesa* and forest landscapes are among those which most require these improvements in information relating to the contribution of nature to the national income of society.

**Supplementary Materials:** The following are available online at www.mdpi.com/1999-4907/12/5/638/s1: Text S1. Accounting frameworks; Text S1.1 Refined standard System of National Accounts; Text S1.2 Comparing the refined SNA and AAS frameworks; Table S1. Livestock units by predominant vegetation type in case-study *dehesa* and forest farms in Andalusia (2010); Table S2. Grazed fodder consumption by species in privately-owned *dehesa* farm in Andalusia (2010: FU/ha); Table S3. Grazed fodder consumption by species in publicly-owned *dehesa* farm in Andalusia (2010: FU/ha); Table S4. Grazed fodder consumption by species in privately-owned forest farm in Andalusia (2010: FU/ha); Table S5. Grazed fodder consumption by species in publicly-owned forest farm in Andalusia (2010: FU/ha); Table S6. Summary of production, income generation, accumulation and capital balance of AAS applied to privately-owned *dehesa* farm with livestock presence in Andalusia (2010: €/ha); Table S7. Summary of production, income generation, accumulation and capital balance of AAS applied to publicly-owned *dehesa* farm with livestock presence in Andalusia (2010: €/ha); Table S8. Summary of production, income generation, accumulation and capital balance of AAS applied to publicly-owned forest farm with livestock presence in Andalusia (2010: €/ha); Table S9. Summary of production, income generation, accumulation and capital balance of AAS applied to privately-owned forest farm with livestock presence in Andalusia (2010: €/ha); Table S10. Production account by activity under AAS applied to privately-owned *dehesa* farm with livestock presence in Andalusia (2010: €/ha); Table S11. Production account by activity under AAS applied to publicly-owned *dehesa* farm with livestock presence in Andalusia (2010: €/ha); Table S12. Production account by activity under AAS applied to privately-owned forest farm with livestock presence in Andalusia (2010: €/ha); Table S13. Production account by activity under AAS applied to publicly-owned forest farm with livestock presence in Andalusia (2010: €/ha); Table S14. Summary of production, income generation, accumulation and capital balance of rSNA applied to privately-owned *dehesa* farm with livestock presence in Andalusia (2010: €/ha); Table S15. Summary of production, income generation, accumulation and capital balance of rSNA applied to publicly-owned *dehesa* farm with livestock presence in Andalusia (2010: €/ha); Table S16. Summary of production, income generation, accumulation and capital balance of rSNA applied to privately-owned forest farm with livestock presence in Andalusia (2010: €/ha); Table S17. Summary of production, income generation, accumulation and capital balance of rSNA applied to publicly-owned forest farm with livestock presence in Andalusia (2010: €/ha); Table S18. Capital balance by activity under AAS applied to privately-owned *dehesa* farm with livestock presence in Andalusia (2010: €/ha); Table S19. Capital balance by activity under AAS for publicly-owned *dehesa* farm with livestock presence in Andalusia (2010: €/ha); Table S20. Capital balance by activity under AAS applied to privately-owned forest farm with livestock presence in Andalusia (2010: €/ha); Table S21. Capital balance by activity under AAS applied to publicly-owned forest farm with livestock presence in Andalusia (2010: €/ha); Table S22. Immobilized capital by activity under AAS applied to privately-owned *dehesa* farm with livestock presence in Andalusia (2010: €/ha); Table S23. Immobilized capital by activity under AAS applied to publicly-owned *dehesa* farm with livestock presence in Andalusia (2010: €/ha); Table S24. Immobilized capital of AAS by activity under AAS

applied to privately-owned forest farm with livestock presence in Andalusia (2010: €/ha); Table S25. Immobilized capital by activity under AAS applied to publicly-owned forest farm with livestock presence in Andalusia (2010: €/ha); Table S26. Immobilized capital of the case-study large *dehesa* and forest farms in Andalusia under refined System of National Account (rSNA) (2010: €/ha).

**Author Contributions:** Conceptualization, P.C.; data curation, P.C., B.M. and A.Á.; formal analysis, P.C., B.M. and A.Á.; funding acquisition, P.C.; methodology, P.C.; project administration, P.C.; supervision, P.C.; visualization, P.C. and B.M.; writing—original draft, P.C., B.M.; writing—review and editing, P.C., B.M. and A.Á. All authors have read and agreed to the published version of the manuscript.

**Funding:** This study was funded by the Agency for Water and Environment of the Regional Government of Andalusia, Contract NET 165602 and the Spanish National Research Council (CSIC), grant number ref. 201810E036.

**Data Availability Statement:** The data presented in this study are available on request from the corresponding author. The data are not publicly available due to ownership rights.

**Acknowledgments:** The authors thank the Agency for Water and Environment of the Regional Government of Andalusia for the financial and field work support for the Renta y Capital de los Montes de Andalucía (RECAMAN) project (Contract NET 165602) and the Valoraciones de Servicios y Activos de Amenidades Privadas de Fincas Silvopastorales (VAMSIL) project of CSIC (ref. 201810E036). We acknowledge the contributions of Alejandro Caparrós, José L. Oviedo, Paola Ovando, Eloy Almazán and other colleagues in the framework of the RECAMAN project to the methods and results presented in this study. We thank Adam B. Collins for helping us to review the English writing.

**Conflicts of Interest:** The authors declare no conflicts of interest.

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
