# Peer review of "Uncovering the Hidden Numbers of Nature in the Standard Accounts of Society: Application to a Case Study of Oak Woodland dehesa and Conifer Forest Farms in Andalusia-Spain"

_forests, doi:10.3390/f12050638_

Round 1

Reviewer 1 Report

Major point

The theoretical discussion in the manuscript is confusing and ambiguous.  

Generally, WTP is the maximum amount of money which the consumer pays for purchasing a good. When the consumer buys it, his/her WTP is usually higher than market price because WTP includes consumer surplus which is defined by the difference of WTP and market price.

Based on this, it seems that the simulation of WTP for the goods without market in the manuscript, cannot be compared with market price directly. In the current manuscript, the explanation is not enough.

In addition, no explanation of stated and revealed preference to estimate the WTP. It only appears in the abstract. The estimating methodology should be introduced and discussed in the main body because how to estimate WTP is quite ambiguous.

Minor points

  • Current abstract is too long. It should be concise.
  • Clarify the notation of TC in equation (1). It’s missing.
  • Overall, the manuscript has redundant and diffuse expressions. Using brief and clear sentence are preferable.

Author Response

REVIEWER 1

Major point

[Reviewer´comment 1]

The theoretical discussion in the manuscript is confusing and ambiguous.  

Generally, WTP is the maximum amount of money which the consumer pays for purchasing a good. When the consumer buys it, his/her WTP is usually higher than market price because WTP includes consumer surplus which is defined by the difference of WTP and market price.

Based on this, it seems that the simulation of WTP for the goods without market in the manuscript, cannot be compared with market price directly. In the current manuscript, the explanation is not enough.

In addition, no explanation of stated and revealed preference to estimate the WTP. It only appears in the abstract. The estimating methodology should be introduced and discussed in the main body because how to estimate WTP is quite ambiguous.

[Authors´response 1]

Our valuations of final products are all at observed market prices, imputed transactions prices (carbon and water) and simulated market transactions prices (unit simulated exchange value). In no case do our valuations include consumer surplus. These are the same valuation criteria as those recommended for valuing ecosystem services and environmental assets in the System of Environmental-Economic Accounting - Ecosystem Accounting (SEEA EA) (UNSD, 2021) [1].

We have incorporated a new paragraph in subsection 3.2.1 lines 365-384 where the simulated exchange value is defined.

Line 11: we change “consumer marginal willingness to pay (WTP)” to “consumer marginal willingness to pay (MWTP)”.

Lines 12-13: we change “simulated market prices” to “simulated exchange values (SEV)”

Line 16: we change “consumer WTP” to “consumer MWTP”.

Lines 16-17: we changeThe simulated exchange values of the private owner and public consumer WTP for these products” to The SEV of private owners and public consumer MWTP for these non-market products”.

Line 43. We change “net value added” to “final products consumed”.

Line 43-44. We change “woody” to “timber, cork, firewood”

Line 44. We change “non-woody” to “non-wood”

We have separated the previous section 2 into the new sections 2 and 3.

Section 3 incorporates accounting framework concepts and methods. We have moved the previous subsections 2.2.1 and 2.2.2 to the Supplementary text S1.

Lines 242-243. We have added the references to the publications by the authors in which the concepts of the activities, economic variables and accounting frameworks are developed in detail. We have also included references to the modifications in Tables 3-9 which present the accounting identities of the individual and aggregate variables for the farmer and government institutional sectors, average farm, the wood and non-wood forest activities as well as the market and non-market products.

Lines 373-382 [revised manuscript Subsection 3.2.1. ]: We have added the paragraph: “ We have valued four classes of final product consumed using consumer stated preference methods (contingent valuation and choice experiment) for estimating the simulated exchange values (marginal willingness to pay) of private amenity, public recreation, landscape and threatened wild biodiversity services. The final product consumed of water has been estimated according to its transaction prices, applying an assumed competitive real discount rate of 3% to the value of the environmental asset of water, estimated using the hedonic prices method [25]. The water transaction price coincides with the environmental price due to the absence of manufactured costs in the production function of water from the case-study farms stored further down the watershed in public reservoirs and used in irrigated croplands.”

Lines 361-418: The previous version of the article Subsection 2.2.3 ‘Product prices under the rSNA and AAS frameworks’ has been refined and extended in the new revised version of the article Subsection 3.2.1.

Line 382-384. In the new version of the article Sub-Section 3.2.1 we include “for valuation methods see reference [20], Sub-section 2.4. Forest Product Valuations, pp. 222-224 and Fig. 2. Methods applied to value forest products in Andalusia, p. 223”.

Line 688-700: We have added “The rates of profitability (return) of wood and non-wood forest products at social prices estimated by the AAS include the manufactured net operating margins (profits) of the non-commercial intermediate product of services (ISSnc) and their counterparts of own ordinary non- commercial intermediate consumption of services (SSncoo). In addition to the abovementioned extension of rSNA, the AAS incorporates wood and non-wood environmental natural growth in total products, that is, of wood and game products less investment consumption of environmental fixed asset of carbon released. The rSNA does not incorporate NG or carbon activity in the measurement of net operating surplus at basic prices. The farmer total profitability rate at social prices in the rSNA does not include the private amenity environmental net operating surplus. Meanwhile, the private amenity EFA in the rSNA is included in the measurement of total land market price. Due to these limitations of the standard SNA it is not possible to compare the total profitability rates of the farmer activities measured by the AAS and SNA.”

Line 704-714: We have added: “In this research, the slight refinement of the standard SNA consists of registering the commercial intermediate product of grazing, separating the net mixed income (NMI) of the individual activity into imputed compensation of unpaid (self-employed) labour and net operating surplus, subtracting the environmental work-in-progress used from the net operating surplus and incorporating it in the intermediate consumption of the individual activity. In addition, the government compensation (operating subsides net of taxes on production) is incorporated as non-commercial intermediate product of service (ISSncc), and their counterpart of ordinary own non-commercial intermediate consumption of service (SSncoo). These changes are intended to make the intermediate product, intermediate consumption and labour compensation under the rSNA and AAS consistently comparable “.

Line 715-723: We have added “The rSNA estimates the total capital of the farmer at market prices, implicitly including the environmental fixed assets of the private amenity in the land market value. The government activities valued by the rSNA include the manufactured capital at market prices of fire services, recreation services, landscape conservation services and threatened wild biodiversity preservation services. The only government activities for which the rSNA estimates the environmental fixed asset values are those of mushrooms and runoff (water yield) with economic use in irrigated land further down the watershed. The rSNA omits the carbon activity in the woodland and forest farms case study”.

Line 834-839: We have added “In all the case-study dehesa and forest farms measured under the AAS, market final products consumed are minor products ranging from 6% lower bound to 29% upper bound of final product consumption in publicly-owned forest farms and privately-owned dehesa farms, respectively (Table 3). The net value added of wood products of the forest only contributes 1.2% lower bound to 8.6% upper bound to the net value added of the farm in the privately-owned forest farm and privately-owned dehesa farm, respectively (Table 4).”

Line 910-918: We have added “Ecosystem services of wood products of the forest do not contribute to the final product consumed of forest farms. While the ecosystem services of wood products of the forest contribute only small percentages of 3.6% and 5.0% to the final products consumed from publicly and privately owned dehesa farms, respectively (Tables 3 and 5). In relative terms, the ecosystem services of wood products account for 10.6% and 5.0% of the ecosystem services of private and public dehesa farms, respectively (Table 5). In these farms, it is cork harvesting that explains these limited contributions of ecosystem services of wood products to the final products consumed and ecosystem services of all activities on the dehesa farms.”

Line 1094-1104. We have added: “The operating profitability rates (Po) of the wood products measured by the AAS vary between the lower bound of 0.9% and the upper bound of 1.8% in the public forest farm and the private dehesa, respectively (Table 8). The total profitability rates (P) of the wood products are significantly higher than those of the Po, with the lower bound of 4.4 % and the upper bound of 8.7 % in the private forest and private dehesa, respectively (Table 8).

As for the economic activities of the farms as a whole, the AAS measures operating profitability rates of 2% at the lower bound and 4.9% at the upper bound for the public and private forest farms, respectively (Table 8). Total farm profitability rates vary between a lower bound of 0.7% and an upper bound of 2.2% for public forest and private dehesa farms (Table 8).”

Line 1150-1165. We have added: “The operating profitability rates for the wood products of the privately-owned dehesa farm estimated by the rSNA and AAS coincide, while in the case of the publicly-owned forest farm the AAS estimates are significantly higher (Table 8). The total profitability rates for wood products of privately-owned dehesa farms estimated by the rSNA and AAS coincide, while for the publicly-owned forest farm the AAS estimates are slightly higher (Table 8).

For all the activities of the case-study farms, the operating profitability rates under the AAS for the privately-owned dehesa significantly exceed those estimated by the rSNA (Table 8). In contrast, the total profitability rates for activities measured by the AAS are significantly lower than those estimated by the rSNA (Table 8).

The reason for the difference in the operating profitability rate measurements under the AAS and rSNA for privately-owned dehesa farms is due to the incorporation of the environmental margin of the private amenity in the AAS. The total profitability rate for the publicly-owned forest farm estimated under the rSNA is different from that estimated by the AAS due to the inclusion of own ordinary non-commercial intermediate consumption of services in the AAS (Table 8).”

Line 1427-1439. We have added “The incorporation of the environmental asset of private amenity is recognised in Spain under land law, which in the case of government purchase of a dehesa or forest allows the payment of a maximum value of twice the present discounted value (NPV) of the resource rents (economic rents) of the market products belonging to the landowner [33,34]. The problem with estimating a fair value for government purchase of rural land is that the government's subjective choice of the coefficient that multiplies the NPV is a source of potential bias in the offer price determined by the government. The government should make use of algorithms generated by economic science to determine through hedonic and/or landowner-stated methods the fair value of the private amenity environmental asset of the farms. In this research this bias has been overcome by estimating the environmental asset of private amenity through the contingent valuation method applied to private non-industrial land owners of dehesa and forest farms in Andalusia [35].”

Line 1473-1481. We have added “This research has demonstrated that the available scientific knowledge allows for consistent measurements of ecosystem services and changes in environmental assets. The SEEA Ecosystem Accounting (EA) methodology agrees with and recommends the principles of valuations based on observed and simulated exchange values of final products consumed with and without market prices. This research values, beyond the SEEA EA recommendations, environmental incomes which integrate ecosystem services and changes in environmental assets in a single variable in a way that is consistent with SEEA EA valuations. Future agreements on the standard monetary SEEA EA must overcome the challenge of incorporating an indicator of environmental income.”

Minor points

[Reviewer´comment 2]

Clarify the notation of TC in equation (1). It’s missing.

[Authors´response 2]

Line 685-687: We have added: “The total cost (TC) comprises intermediate consumption (IC), labour (LC) and consumption of fixed capital (CFC) (see details in [13], SM S1, p. 3., SE9)”.

[Reviewer´comment 3]

Current abstract is too long. It should be concise.

[Authors´response 3]

We have reduced the abstract and introduction sections.

The editor of the special issue Dr. Mario Soliño invited us to submit our paper in the context of our two previous articles, the novelty in this case being that this new article is a micro scale case study of dehesas and forest farms. The aim of our paper is to present a complete comparison of multiple woodlands and forests. This comparison takes into account the income and capital valuations of market and non- market products. In addition, ownership rights play an important role in the economic results of the farm case study. The shortcomings of standard System of National Accounts and the omission of the Ecosystem Accounting (SEEA EA) recommendation of the valuation principles are issues covered in our discussion and conclusion. We prefer to retain the complete explanation of this novel issue, as is the environmental income, with the aim of improving the ongoing monetary SEEA EA. These are the reasons for which Dr. Mario Soliño agreed that, in our case, a paper of this length is acceptable on this occasion.

The increased length of this new version is due to the inclusion of the concepts and methods in the main text as proposed by the reviewer. In the previous version of the article these concepts and methods were presented in Supplementary texts S1 and S2.

[Reviewer´comment 4]

Overall, the manuscript has redundant and diffuse expressions. Using brief and clear sentence are preferable.

[Authors´response 4]

We have reviewed the text and believe that the latest changes have improved the content and clarity of the sentences. However, this paper addresses multiple issues which need to be dealt with in sufficient detail to embrace their full significance.

Reviewer 2 Report

Dear Authors
I suggest to reduce the length of article, because the readability is dramatically low. You can express the concepts in shorter way.
Moreover, the results reported the application of two methods to assess the value of a specific typology of environment, which increase the knowledge on the topic, but a scientific reader would appreciate the division between values applicable in the real economy and non-market values in table 3, 4 and 5. I also suggest to group some values according some group categories, like non-wood forest products, wood products, farmed products etc. So far there is an ongoing debate on how classify ecosystem services for reliable estimations, which are approaching a legislative adoption at least at EU level. I suggest to read Fisher & Turner (2008) or La Notte et al. (2017). 
In my option the paper should be improve in order to increase the readability and usability of the interesting data you reported.

Author Response

REVIEWER 2

[Reviewer´ comment 1]

I suggest to reduce the length of article, because the readability is dramatically low. You can express the concepts in shorter way.

[Authors´response 1]

The editor of the special issue Dr. Mario Soliño invited us to submit our paper in the context of our two previous articles, the novelty in this case being that this new article is a micro scale case study of dehesas and forest farms. The aim of our paper is to present a complete comparison of multiple woodlands and forests. This comparison takes into account the income and capital valuations of market and non- market products. In addition, ownership rights play an important role in the economic results of the farm case study. The shortcomings of standard System of National Accounts and the omission of the Ecosystem Accounting (SEEA EA) recommendation of the valuation principles are issues covered in our discussion and conclusion. We prefer to retain the complete explanation of this novel issue, as is the environmental income, with the aim of improving the ongoing monetary SEEA EA. These are the reasons for which Dr. Mario Soliño agreed that, in our case, a paper of this length is acceptable on this occasion.

The increased length of this new version is due to the inclusion of the concepts and methods in the main text as proposed by the reviewer. In the previous version of the article these concepts and methods were presented in Supplementary texts S1 and S2.

[Reviewer´ comment 2]

Moreover, the results reported the application of two methods to assess the value of a specific typology of environment, which increase the knowledge on the topic, but a scientific reader would appreciate the division between values applicable in the real economy and non-market values in table 3, 4 and 5. I also suggest to group some values according some group categories, like non-wood forest products, wood products, farmed products etc.

[Authors´response 2]

In Table 3 the final products consumed have been aggregated into market and non-market products. In Tables 4 to 9 we have grouped the individual activities of the institutional sectors of the farmer and the government into wood products and non-wood forest products.

The results associated with these product classifications are described in new texts (see lines 834-839, 910-918, 1094-1104, 1150-1165).

[Reviewer´ comment 3]

So far there is an ongoing debate on how classify ecosystem services for reliable estimations, which are approaching a legislative adoption at least at EU level. I suggest to read Fisher & Turner (2008) or La Notte et al. (2017). 

[Autors´response 3]

Future EU legislation on the classification and valuation of ecosystem services and environmental assets will be based on the standard System of Environmental-Economic Accounting - Ecosystem Accounting (SEEA EA) (UNSD, 2021) [1]. Our article adopts the definitions and principles of ecosystem service and environmental asset valuation recommended by the SEEA EA.

[Reviewer´ comment 4]

In my option the paper should be improve in order to increase the readability and usability of the interesting data you reported.

[Authors´response 4]

We have reduced the abstract and introduction sections.

We have separated the previous section 2 into the new sections 2 and 3.

Section 3 incorporates accounting framework concepts and methods. We have moved the previous subsections 2.2.1 and 2.2.2 to the Supplementary text S1.

Section 3 and 4 have been extended with new paragraphs (lines 373-382, 688-700, 704-714, 715-723), including wood products and non-wood forest products results (lines 834-839, 910-918, 1094-1104, 1150-1165).

Round 2

Reviewer 2 Report

Dear authors

the paper has been slightly improved, but the length and the syntax decrease the readability of your work. If you can, please reduce the length in order to keep the essential information, while all the other tables can be put in annex. This is simply a recommendation because the paper itself can be accepted for the contents and methodology.
Best regards

Author Response

Manuscript ID: forests-1173507

REVIEWER 2

[Reviewer´s comment]

The paper has been slightly improved, but the length and the syntax decrease the readability of your work. If you can, please reduce the length in order to keep the essential information, while all the other tables can be put in annex. This is simply a recommendation because the paper itself can be accepted for the contents and methodology.

[Authors´response]

The Section 3 of "Concepts and methods of the accounting frameworks" is unusually long because it responded in the first revision of the article to the reviewers' request to expand the definitions and identities of the accounting methodologies applied. We consider the large number of words in Section 3 to be appropriate, thus avoiding a return to the first version of the paper where the concepts and methods were presented in summary form.

Although we recognise the difficulty of reading the article because of its length, we nevertheless consider that we should not reduce its size without degrading its extensive and complex content.

We have revised the text in detail and deleted some sentences, although this is only a slight reduction in the number of words.

The above reasons justify that in the interest of the quality of the content we maintain the current size of the article.
